# Resa: Efficient Reasoning Models via SAEs

## Abstract

How cost-effectively can we elicit strong reasoning abilities in language models by leveraging their underlying representations? We present Resa, a family of reasoning models trained via an efficient sparse autoencoder tuning (SAE-Tuning) procedure. This method first trains an SAE to capture reasoning abilities from a source model, and then uses the trained SAE to guide a standard supervised fine-tuning process to elicit such abilities in a target model, all using verified question-answer data *without any reasoning traces*. When applied to certain Qwen-style models before further RL training, SAE-Tuning retains 97% of its RL-trained counterpart's performance while reducing training costs by 2000x to roughly $1 and training time by 450x to around 20 minutes. Furthermore, even at the 1.5B model size, SAE-Tuning on lightly RL-trained models delivers strong reasoning results, reaching 43.33% Pass@1 on AIME24 and 90% Pass@1 on AMC23. We also show that SAE-Tuning works for Llama-style models, boosting their scores by over 10% on tasks like AMC23 and MATH500. Surprisingly, the reasoning abilities extracted via SAEs are potentially both generalizable and modular. Generality means abilities extracted from one dataset still elevate performance on a larger and overlapping corpus. Modularity means abilities extracted from models like Qwen or Qwen-Math can be attached to the R1-Distilled Qwen model at test time, *without any retraining*, and yield comparable gains. Extensive ablations validate these findings and all artifacts are fully open-sourced.

## 1 Introduction

Reasoning language models have demonstrated increasing performance in domains like math, coding, and science (Xu et al., 2025; Liu et al., 2025). Despite the impressive reasoning performance elicited by reinforcement learning (RL) or supervised fine-tuning (SFT) (Chu et al., 2025), these methods often operate as a "black box". In other words, while they improve reasoning, how they alter the model's internal representations to do so is largely opaque. Furthermore, RL-based workflows are notoriously resource-intensive, requiring substantial compute and long training time to converge. SFT, in turn, hinges on the availability of high-quality Chain-of-Thought (CoT) reasoning traces, which are costly to curate (Muennighoff et al., 2025). This leaves a critical gap: The need for a "three-birds-one-stone" method that can elicit strong reasoning abilities in a way that is not only effective but also computationally efficient and transparent.

We aim to bridge this gap with Resa, a family of reasoning models trained via sparse autoencoders (SAEs) on Qwen and Llama models, using a novel post-training method, which we call *SAE-Tuning*. Specifically, SAE-Tuning involves two key stages: First, we use an SAE to probe the internal activations of a source model, extracting a dictionary of latent features that correspond to its reasoning processes. SAEs are unsupervised models designed to deconstruct a model's dense internal activations into a sparse dictionary of more interpretable latent features (Anthropic, 2023; 2024). Our assumption is that within this dictionary, certain features should correspond to the building blocks of reasoning abilities. Second, we freeze this feature-rich SAE and insert it into a target model to guide a SFT process to elicit reasoning abilities in the target model. By instilling latent reasoning features captured by an SAE back into a model via a tuning procedure, we can effectively and efficiently elicit the model's reasoning abilities.

SAE-Tuning also distinguishes itself from existing methods by using SFT on a minimal verified CoT-free question-answer data type. By verified, we mean that the answer correctness is ensured via methods like human annotation or language-model-based verification (Guha et al., 2025), while

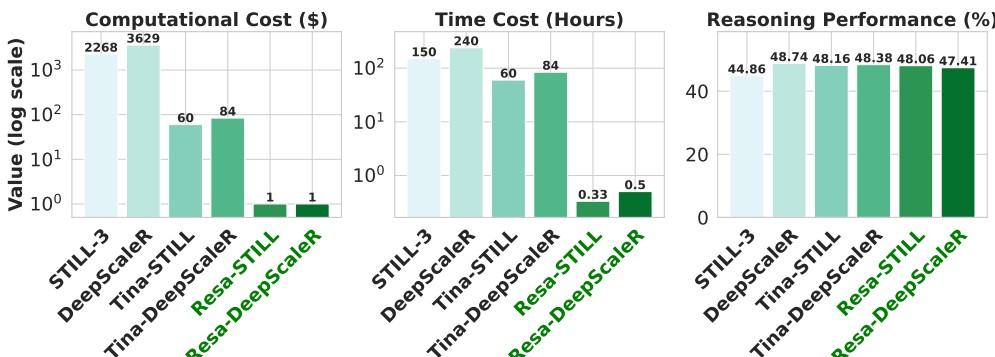

Figure 1: **Comparison of Example Resa Models and Baselines** The Tina models correspond to the best checkpoints in Wang et al. (2025a). Resa-STILL and Resa-DeepScaleR correspond to Resa-STILL-v5 and Resa-DeepScaleR-v3 in Table 2, respectively. For these Resa models, the required SAEs are trained from scratch (as shown in Section 4.1) and both computational and time costs are total costs for training SAEs and models. Reasoning performance denotes the average zero-shot Pass@1 score across AIME24/25, AMC23, MATH500, GPQA Diamond, and Minerva benchmarks. Please see more details in the Appendix B.1.

CoT-free signifies that our SAE-Tuning procedure functions without needing explicit step-by-step reasoning traces[1]. In line with this minimalist approach, we demonstrate our method across multiple model families and sizes, focusing on Qwen and Llama architectures at the 1.5B and 3B scale. This strategic choice lowers computational costs—thereby democratizing research—and allows us to more precisely isolate and measure the incremental benefits of SAE-Tuning. We summarize our core contributions as follows:

- **Efficient Reasoning Ability Elicitation** Purely using verified CoT-free data, we demonstrate that SAE-Tuning can be applied in an end-to-end manner to certain base models with a trained-from-scratch SAE to elicit reasoning abilities on par with those achieved via costly RL. This leads to substantial gains with peak training cost reductions of over 2000x (to approximately $1) and time reductions of over 450x (to under 20 minutes) compared to RL-based workflows, while maintaining comparable performance.

- **Generalizable and Modular Reasoning Ability** We establish the generality and modularity of the extracted reasoning abilities such that these abilities generalize across out-of-distribution datasets and can be attached to models within the same family at test time without additional training, functioning as a portable reasoning adapter.

## 2 RELATED WORK

**Reinforcement Learning for Reasoning Ability Elicitation** The structure of reasoning tasks lends itself well to RL approaches, primarily because the final output's correctness provides a clear and verifiable reward signal. This feedback loop helps the model develop more robust reasoning strategies (Shao et al., 2024; DeepSeek-AI, 2025). Recently, a growing body of work suggests that RL primarily elicits and amplifies reasoning capabilities already embedded within pretrained models, rather than installing them from scratch. Training dynamics analysis supports this "elicitation hypothesis," showing that post-training largely surfaces latent abilities (Zhao et al., 2025). The elicitation hypothesis is substantiated by several findings. For example, significant reasoning gains are achievable through minimal parameter updates that merely teach the model a new output format (Wang et al., 2025a), and even through one-shot RL with data selection (Wang et al., 2025b). More surprisingly, studies have shown that RL can surface reasoning skills even with spurious or incorrect rewards (Shao et al., 2025), indicating that the primary mechanism is the surfacing of useful, pre-existing representations. We in this paper show that one can perform such elicitation in a much more efficient way by bypassing RL.

---

[1]Results in Table 2 show that standard SFT on CoT-free data without an SAE fails to elicit reasoning.

**Sparse Autoencoders** Recent advances in SAEs have enabled new approaches for analyzing and steering neural network computations. Building on the original SAE architecture proposed by Cunningham et al. (2023), subsequent work from Anthropic (2023) demonstrated how these sparse bottleneck networks can decompose transformer activations into human-interpretable features. The scaling properties of SAEs were systematically studied in Anthropic (2024), establishing practical guidelines for training SAEs across model sizes. Recent innovations have improved SAE training stability and feature quality: O'Neill & Bui (2024) introduced scalable and reliable circuit identification techniques. Concurrent work by Chen et al. (2025) has focused on optimizing SAE computational efficiency and integration with modern transformer architectures. Karvonen et al. (2025) developed a comprehensive evaluation on components of SAE training. In the SAE-Tuning procedure, we leverage SAEs specifically to extract the latent features that underpin reasoning abilities.

**Model Steering** The use of SAEs for model steering builds on earlier work in activation editing (Alain & Bengio, 2018). Panickssery et al. (2024) first demonstrated that SAE features could be used for controlled behavior modification, while Bayat et al. (2025) later developed more precise steering vectors through feature subspace analysis. Recent work by O'Brien et al. (2025) demonstrated improved safety properties through SAE-mediated interventions and Gan et al. (2025) showing SAEs' transferability of steering across modalities. Alternative to SAEs, activation differences (Li et al., 2023) and recursive feature machines (Beaglehole et al., 2025) are also widely used for steering, and Wu et al. (2025) evaluated the concept steering abilities of these methods. Besides steering, Chen et al. (2025) demonstrates the feasibility of adapting models to pretrained SAEs. Our proposed procedure, SAE-Tuning, beyond steering and adapting, fully leverages sparse autoencoders to identify, extract, and elicit latent reasoning abilities.

## 3 RESA: EFFICIENT REASONING MODELS VIA SAEs

Resa is a family of reasoning models derived from the Qwen (Section 4.1) and Llama (Section 4.2) families of models. We use an SAE to explicitly isolate and extract implicit reasoning abilities (i.e., latent reasoning features) from a source model, and use this trained SAE to controllably instill those features into a target model to elicit reasoning abilities. We refer to this post-training procedure as SAE-Tuning, which requires only verified CoT-free question-answer data.

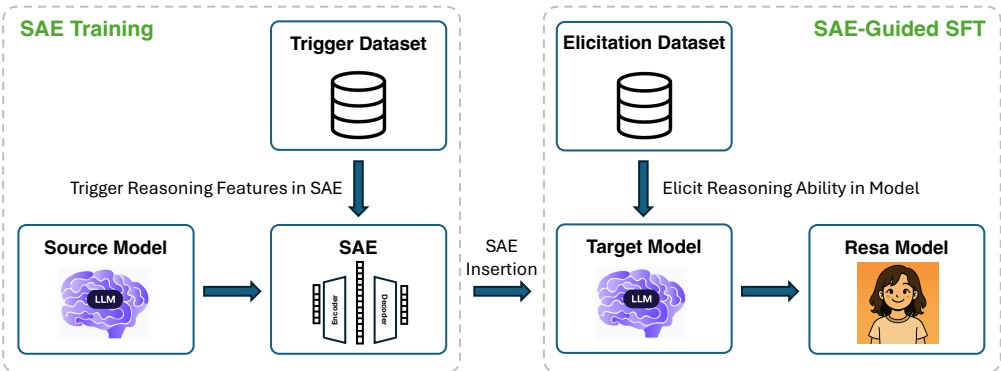

Figure 2: **SAE-Tuning: A Two-Stage Pipeline** The procedure begins with SAE training (Left), where an SAE is trained to capture reasoning features from a source model with a trigger dataset. During SAE-guided SFT (Right), the trained SAE is then frozen and inserted into a target model. An elicitation dataset is used to guide the SFT to elicit the reasoning abilities in the target model.

For the primary end-to-end configuration in Figure 2, the source and target models are the same base model, and the trigger and elicitation datasets are the same CoT-free data. Our experiments also explore configurations where the source model, target model, and datasets differ (e.g., for ability replication in Appendix C.1 and cross-dataset transfer in Section 5). A detailed configuration of all main Resa models is provided in Table 1.

| MAIN MODEL | SOURCE MODEL | TARGET MODEL | TRIGGER DATA | ELICITATION DATA | SAE TRAINING MODE |
|---|---|---|---|---|---|
| **Resa-STILL-v1 (in Table 7)** | Tina-STILL | R1-Distill | STILL | STILL | Fine-tuned |
| **Resa-STILL-v2 (in Table 2)** | - | R1-Distill | - | STILL | Pre-trained |
| **Resa-STILL-v3 (in Table 2)** | Tina-STILL | R1-Distill | STILL | STILL | Trained-from-Scratch |
| **Resa-STILL-v4 (in Table 2)** | R1-Distill | R1-Distill | STILL | STILL | Fine-tuned |
| **Resa-STILL-v5 (in Table 2)** | **R1-Distill** | **R1-Distill** | **STILL** | **STILL** | **Trained-from-Scratch** |
| **Resa-DeepScaleR-v1 (in Table 7)** | Tina-DeepScaleR | R1-Distill | DeepScaleR | DeepScaleR | Fine-tuned |
| **Resa-DeepScaleR-v2 (in Table 2)** | Tina-DeepScaleR | R1-Distill | DeepScaleR | DeepScaleR | Trained-from-Scratch |
| **Resa-DeepScaleR-v3 (in Table 2)** | **R1-Distill** | **R1-Distill** | **DeepScaleR** | **DeepScaleR** | **Trained-from-Scratch** |
| **Resa-DeepScaleR-v4 (in Table 4)** | Tina-STILL | R1-Distill | STILL | DeepScaleR | Fine-tuned |

Table 1: **Main Resa Models** Each row outlines the components used in SAE-Tuning. The bold rows represent the key configuration where reasoning abilities are extracted from the base model itself and then instilled back into the same model, i.e., the source and target models are the same. R1-Distill is the 1.5B distilled Qwen model (DeepSeek-AI, 2025) and Tina models are further RL-trained models on R1-Distill. (Wang et al., 2025a)

## 3.1 SPARSE AUTOENCODER TUNING: WORKFLOW & INTUITION

SAE-Tuning is an efficient two-stage training procedure to transfer reasoning abilities from a source model to a target model; this procedure is summarized in Figure 2. The two stages consist of:

**Stage I: SAE Training (Reasoning Ability Extraction)** The first stage involves training an SAE to reconstruct the activations from a specific layer of a source model. We feed a trigger dataset, comprising only verified CoT-free question-answer pairs, into the source model and capture the resulting activations at a chosen SAE hookpoint. The SAE is then trained on these activations, learning features that represent the source model's internal reasoning while processing the data.

**Stage II: SAE-Guided SFT (Reasoning Ability Elicitation)** Once the SAE has been trained, we shift to training a target model. The trained SAE is actively integrated into the target model's architecture (at certain layer) and kept with frozen weights during a standard SFT process. By exposing the target model to the feature representations captured by the SAE, the SFT process is guided to develop internal pathways that elicit reasoning abilities, effectively reconstructing such abilities extracted from the source model. This entire stage uses an elicitation dataset, which is typically identical to the trigger dataset.

As shown in the following, the SAE-Tuning procedure is configured by five key components.

- **Source Model** The model from which reasoning-related features are extracted. The SAE is trained on the intermediate activations of one of its layers.

- **Target Model** The model for which we aim to elicit the reasoning abilities.

- **Trigger Dataset** The CoT-free dataset used to trigger reasoning-related features in SAEs during SAE training. It is constructed from a standard open-source question-answer dataset by simply formatting each entry into a specific template. For a given question and its final answer, we have an input sequence: Problem: [Question] `<think>` [Answer] `</think>` `<answer>` Answer: [Answer] `</answer>`. While this template uses `<think>` and `</think>` tokens, the dataset remains CoT-free as no intermediate reasoning steps are provided between the tokens, only the answer is present. The inclusion of this structure is hypothesized to activate the source model's latent reasoning abilities, allowing the SAE to capture such features. A detailed analysis of the role and importance of these thinking tokens is provided in Section C.2.

- **Elicitation Dataset** The CoT-free dataset used for the SAE-guided SFT of the target model to elicit reasoning abilities. In our experiments, it is usually the same as the trigger dataset.

- **SAE Training Mode** This defines how Stage I (i.e., SAE training) is carried out. We explore three distinct modes. (1) Pre-trained: We use an SAE that has been pre-trained on general pre-training text corpus, e.g., SAEs on R1-Distill.[2] This mode bypasses Stage I entirely. (2) Fine-tuned: The pre-trained SAE is further fine-tuned on activations from the source model using

---

[2]EleutherAI/sae-DeepSeek-R1-Distill-Qwen-1.5B-65k

the trigger dataset. (3) Trained-from-Scratch: The SAE is trained from a random initialization, exclusively on the activations produced by the source model with the trigger dataset.

At its core, SAE-Tuning operates on a simple principle, namely:

> Enforcing a reasoning-aligned structure on a model's internal representations.

During Stage I, the autoencoder observes a capable source model and learns to deconstruct its complex internal activations into a sparse dictionary of features. Our central hypothesis is that a subset of these features represents the fundamental building blocks of reasoning. In Stage II, this feature dictionary acts as a fixed "template." By inserting the frozen SAE into a target model and fine-tuning the model to make its internal activations compatible with this template, we constrain its learning process. The target model is thus guided to arrange its internal representations to mimic the effective reasoning structures captured from the source model, directly instilling those abilities into its own parameters. We also offer two more alternative perspectives to build intuition for SAE-Tuning.

**Intuition I: Knowledge Distillation** The SAE acts as a "knowledge bottleneck." In Stage I, it is forced to learn a compressed and essential representation of the source model's reasoning processes. In Stage II, it becomes a "teacher." The KL divergence objective distills this knowledge into the target (student) model's parameters, effectively teaching the student model to replicate the teacher's reasoning behavior while being guided by the explicit reasoning features captured by the SAE. We provide more detailed discussion in Section 4.1.

**Intuition II: Alternating Optimization** We are optimizing two distinct sets of parameters for two different goals, one after the other. 1) Optimizing the SAE: We hold the model constant and train the SAE parameters to best capture the model's latent reasoning features. 2) Optimizing Model Adapters: We hold the model and the SAE constant and train low-rank adapters to best integrate the SAE's reasoning feature representation into the model. Combining these two goals allows the SAE to act as a bridge between the two models. The first optimization step builds this bridge by learning a compressed blueprint of the source model's reasoning process. The second step then fine-tunes the target model align with the source model, ensuring that the target model inherits the structural properties of the source model's reasoning.

## 3.2 SPARSE AUTOENCODER TUNING: MATHEMATICAL FORMULATION

We now formalize the two primary stages of the SAE-Tuning procedure.

**Stage I: SAE Training** Given a source model $M_0$, we denote the SAE to be trained as $s_\ell$, which is hooked at $\ell$-th layer (i.e., the multilayer perceptron output) of the source model $M_0$. Suppose the source model has $L$ layers with hidden dimension $d$. Given input $\mathbf{x}_0$ and output $\mathbf{y}$, we denote the activation after the $\ell$th layer by $\mathbf{x}_\ell$. We view the $\ell$th transformer block as a function $h_\ell$ and we have

$$\mathbf{x}_\ell = h_\ell(\mathbf{x}_{\ell-1}), \quad 1 \le \ell \le L, \tag{1}$$
$$\mathbf{y} = \mathrm{softmax}(\mathbf{x}_L). \tag{2}$$

The SAE $s_\ell$ trains an encoder $\mathbf{W}_{\mathrm{enc}} \in \mathbb{R}^{m \times d}$ for $m \gg d$, a decoder $\mathbf{W}_{\mathrm{dec}} \in \mathbb{R}^{d \times m}$ with unit norm columns, and biases $\mathbf{b}_{\mathrm{enc}} \in \mathbb{R}^m$, $\mathbf{b}_{\mathrm{dec}} \in \mathbb{R}^d$. For activation $\mathbf{x}_\ell$, the SAE reconstructs activation $\tilde{\mathbf{x}}_\ell$ as

$$\mathbf{z} = \text{Top-}k(\mathbf{W}_{\mathrm{enc}}(\mathbf{x}_\ell - \mathbf{b}_{\mathrm{dec}}) + \mathbf{b}_{\mathrm{enc}}), \tag{3}$$
$$\tilde{\mathbf{x}}_\ell = \mathbf{W}_{\mathrm{dec}}\mathbf{z} + \mathbf{b}_{\mathrm{dec}} = \sum w_i f_i, \tag{4}$$

where Top-$k$ means that we only keep the top $k$ features in the vector (Gao et al., 2024). This is a simple and standard practice when training SAEs. The SAE minimizes the reconstruction error

$$\mathcal{L} = \|\mathbf{x}_\ell - \tilde{\mathbf{x}}_\ell\|^2. \tag{5}$$

**Stage II: SAE-Guided SFT** For the trained SAE $s_\ell$ with a hookpoint $\ell$, we freeze its weights and insert it[3] immediately after layer $\ell$ of a target model $M$. Note that this operation requires the source and target models to have the same underlying model architecture and size such that the SAE can be inserted directly. We denote the intermediate activation after $i$-th layer, before and after SAE

---

[3]We only consider single-layer SAE insertion, i.e., insert at most one SAE at a time.

insertion, as $\mathbf{x}_i$ and $\tilde{\mathbf{x}}_i$, respectively. Given the SAE $s_\ell$ with a hookpoint $\ell$, we have $\tilde{\mathbf{x}}_i = \mathbf{x}_i, i \leq \ell - 1$ and the reconstructed activation $\tilde{\mathbf{x}}_\ell = \text{SAE}(\mathbf{x}_\ell)$ propagates through the remaining layers to produce

$$\tilde{\mathbf{x}}_i = h_i(\tilde{\mathbf{x}}_{i-1}), \quad \ell + 1 \leq i \leq L, \tag{6}$$

$$\tilde{\mathbf{y}} = \text{softmax}(\tilde{\mathbf{x}}_L). \tag{7}$$

We then add low-rank adapters of rank $r$ in each multilayer perceptron and attention sublayer of every layer of the target model we are adapting. We provide insights on why we choose low-rank adapters in Section 5. Concretely, for each frozen weight matrix $W_i \in \mathbb{R}^{d_1 \times d_2}$, we add $A_i \in \mathbb{R}^{d_1 \times r}$ and $B_i \in \mathbb{R}^{r \times d_2}$. We train only the low-rank adapters $\Theta = \{A_i\} \cup \{B_i\}$. The objective is the KL divergence between the next token probability distribution with and without the SAE inserted:

$$\underset{\Theta}{\arg\min} \, \mathcal{D}_{\text{KL}}(\tilde{\mathbf{y}}, \mathbf{y}). \tag{8}$$

The core intuition behind this loss function is to force the target model to produce internal representations at $\ell$-th layer that are "compatible" with the frozen SAE with rich reasoning features. Since the SAE was trained to reconstruct the reasoning-focused activations of the source model, this objective pushes the target model's activations to become explicitly similar to the source model's internal reasoning structure. By fine-tuning the target model to accommodate the SAE with minimal disruption to its output, we instill the reasoning patterns embodied by the SAE's learned features.

**Inference Stage** Crucially, after this SAE-guided SFT is complete, the SAE is entirely removed from the model at test time. This leaves an enhanced target model with the elicited reasoning abilities "instilled" directly into its own parameters, ready for standard inference and evaluation.

## 4 EFFICIENT REASONING ABILITY ELICITATION

We show the primary practical utility of SAE-Tuning that it is effective when the source and target models are the same, thus bypassing the need for further RL. We also show the method's self-sufficiency with a trained-from-scratch SAE, which eliminates the dependence on pre-trained SAEs.

**Training Setup** The default configuration for SAE-Tuning is as follows. The primary datasets are STILL[4] (RUCAIBox STILL Team, 2025) and DeepScaleR[5] (Luo et al., 2025). By default, we set the source and target models as R1-Distill. The SAE is hooked to the output of the multilayer perceptron submodule after the 12th layer (out of 28) of the source model. This choice is based on the heuristic that middle layers in a transformer-based language model are often crucial for understanding and reasoning. We provide a detailed discussion on layer selection in Section C.2. The full hyperparameter setting is provided in Appendix B.2.

**Evaluation Setup** All evaluations reported herein utilize the `lighteval` framework (Habib et al., 2023) integrated with the `vLLM` (Kwon et al., 2023) inference engine for efficiency. We maintain a fixed hardware configuration (two GPUs) and apply a standardized set of `vLLM` inference parameters across all evaluated models. Specifically, the temperature is set as 0.6 and the top-p value is 0.95. All scores are zero-shot Pass@1 Mean@10 performance. Particularly, we evaluate the reasoning abilities of models across a diverse suite of six reasoning benchmarks, primarily focused on mathematical and scientific reasoning: AIME24/25 (Art of Problem Solving, 2024), AMC23 (Art of Problem Solving, 2023), MATH500 (Hendrycks et al., 2021; Lightman et al., 2023), Minerva (Lewkowycz et al., 2022), and GPQA Diamond (short as GPQA in this rest of the paper) (Rein et al., 2024).

### 4.1 EXPERIMENTS ON QWEN-STYLE ARCHITECTURES

In this section, we investigate the practical usage of SAE-Tuning that if SAE-Tuning can be done in an end-to-end procedure: if one can elicit reasoning abilities directly from the base model, without a pre-trained SAE. Specifically, the SAE only needs to be trained on top of the trigger dataset, the source and target models are the same.

**SAE Training Mode** The SAE training mode ablation in Table 2 shows that training an SAE from scratch on the trigger dataset (i.e, Resa-STILL-Trained-from-Scratch-SAE, 47.36% avg) is just as

---

[4]RUC-AIBOX/STILL-3-Preview-RL-Data

[5]agentica-org/DeepScaleR-Preview-Dataset

effective as fine-tuning a generic pre-trained SAE on the same dataset (i.e., Resa-STILL-Finetuned-SAE, 47.28% avg). Both outperform using a default, pre-trained SAE (44.99% avg). The key insight is that the SAE's performance for reasoning ability elicitation hinges on its exposure to the specific reasoning features encapsulated in the data (i.e., the trigger dataset), while the general knowledge from its initial pre-training is less critical in SAE-Tuning. This result aligns with the knowledge distillation perspective of SAE-Tuning at the end of Section 3.1. In that view, the SAE is a "teacher" guiding the "student" (i.e., the target model). A static, pre-trained SAE is an ineffective teacher because it is ignorant of the "curriculum"—the reasoning patterns in the trigger dataset. In contrast, both training from scratch and fine-tuning are effective because they ensure the teacher first learns the specific material it is meant to teach. Overall, this finding simplifies the pipeline by eliminating the need for a pre-trained SAE, which yields substantial compute savings by avoiding the costly pre-training on large corpora like SmolLM2 (Allal et al., 2025) and RedPajama (Weber et al., 2024).

| MODEL NAME | AIME24 | AIME25 | AMC23 | MATH500 | GPQA | MINERVA | AVG. |
|---|---|---|---|---|---|---|---|
| STILL-3-1.5B-preview | 26.67 | 26.67 | 67.50 | 86.40 | 34.34 | 27.57 | 44.86 |
| Tina-STILL | 36.67 | 30.00 | 77.50 | 84.60 | 33.33 | 26.84 | 48.16 |
| SAE TRAINING MODE ABLATION | AIME24 | AIME25 | AMC23 | MATH500 | GPQA | MINERVA | AVG. |
| Resa-STILL-Finetuned-SAE (i.e., Resa-STILL-v1) | 33.33 | 33.33 | 75.00 | 83.80 | 29.41 | 28.79 | 47.28 |
| Resa-STILL-Pretrained-SAE (i.e., Resa-STILL-v2) | 23.33 | 23.33 | 72.50 | 85.40 | 30.51 | 34.85 | 44.99 |
| Resa-STILL-Trained-from-Scratch-SAE (i.e., Resa-STILL-v3) | 33.33 | 33.33 | 70.00 | 83.00 | 30.15 | 34.34 | 47.36 |
| SOURCE MODEL ABLATION (FINE-TUNED SAE) | AIME24 | AIME25 | AMC23 | MATH500 | GPQA | MINERVA | AVG. |
| Resa-STILL-Base (i.e., Resa-STILL-v4) | 23.33 | 20.00 | 77.50 | 84.60 | 27.57 | 35.35 | 44.73 |
| Resa-STILL-Tina-1-step | 40.00 | 26.67 | 70.00 | 83.20 | 31.62 | 36.36 | 47.98 |
| Resa-STILL-Tina-10-step | 23.33 | 23.33 | 75.00 | 82.20 | 29.41 | 33.33 | 44.43 |
| Resa-STILL-Tina-50-step | 33.33 | 26.67 | 72.50 | 82.60 | 27.57 | 38.89 | 46.93 |
| Resa-STILL-Tina-100-step | 43.33 | 23.33 | 82.50 | 85.60 | 28.68 | 33.33 | 49.46 |
| Resa-STILL-Tina-500-step | 36.67 | 26.67 | 80.00 | 83.80 | 31.62 | 31.82 | 48.43 |
| Resa-STILL-Best-Tina-2000-step (i.e., Resa-STILL-v1) | 33.33 | 33.33 | 75.00 | 83.80 | 29.41 | 28.79 | 47.28 |
| Resa-STILL-Tina-3000-step | 26.67 | 23.33 | 72.50 | 85.40 | 27.94 | 34.85 | 45.11 |
| SOURCE MODEL ABLATION (TRAINED-FROM-SCRATCH SAE) | AIME24 | AIME25 | AMC23 | MATH500 | GPQA | MINERVA | AVG. |
| Resa-STILL-Base (i.e., Resa-STILL-v5) | 33.33 | 26.67 | 70.00 | 87.00 | 29.41 | 41.92 | 48.06 |
| Resa-STILL-Tina-1-step | 33.33 | 33.33 | 72.50 | 82.20 | 29.41 | 35.35 | 47.69 |
| Resa-STILL-Tina-10-step | 33.33 | 16.67 | 67.50 | 86.20 | 30.51 | 37.37 | 45.26 |
| Resa-STILL-Tina-50-step | 43.33 | 23.33 | 77.50 | 83.40 | 29.41 | 38.89 | 49.31 |
| Resa-STILL-Tina-100-step | 33.33 | 23.33 | 90.00 | 82.60 | 28.68 | 35.35 | 48.88 |
| Resa-STILL-Tina-500-step | 36.67 | 20.00 | 67.50 | 84.20 | 30.88 | 35.35 | 45.77 |
| Resa-STILL-Best-Tina-2000-step (i.e., Resa-STILL-v3) | 33.33 | 33.33 | 70.00 | 83.00 | 30.15 | 34.34 | 47.36 |
| Resa-STILL-Tina-3000-step | 30.00 | 20.00 | 77.50 | 86.20 | 31.62 | 37.88 | 47.20 |
| Resa-DeepScaleR-Best-Tina-1000-step (i.e., Resa-DeepScaleR-v2) | 40.00 | 30.00 | 75.00 | 84.00 | 30.15 | 33.33 | 48.75 |
| Resa-DeepScaleR-Base (i.e., Resa-DeepScaleR-v3) | 33.33 | 23.33 | 80.00 | 86.00 | 30.51 | 31.31 | 47.41 |

Table 2: **SAE-Tuning on Qwen-Style Models** (SAE Training Mode Ablation) Training an SAE from scratch is comparably as effective as fine-tuning a pre-trained SAE. (Source Model Ablations) The key results are Resa-STILL-v5 and Resa-DeepScaleR-v3 models which use the base model as their own source and match the reasoning performance of the RL-trained models. More detailed results are shown in Appendix C.4.

**Source and Target Models** Based on the above experiments on SAEs, we now investigate the source and target model used for SAE training, using models ranging from the base R1-Distill model (i.e., models with the "Base" postfix) to further RL-trained checkpoints on R1-Distill (i.e., models with the "Tina" postfix) (Wang et al., 2025a). We notice that the source of reasoning features is nuanced such that there is a non-monotonic relationship between the source model's training progression and the resulting Resa model's performance. The "best" reasoning features for extraction are not always found in the final, most-trained source model checkpoint. Specifically, it shows that one can get optimal performance with a light RL training, i.e., Resa-STILL-Tina-100-step (with fine-tuned SAE, 49.46%) and Resa-STILL-Tina-50-step (with train-from-scratch SAE, 49.31%). Another important finding is that by training an SAE from scratch and using the base model as the source, our method achieves a competitive average score of 48.06% (i.e., Resa-STILL-v5). This performance is nearly identical to the fully RL-trained Tina-STILL model (48.16%), demonstrating that our simplified, end-to-end SAE-Tuning procedure has the potential to replace the RL fine-tuning stage with no meaningful loss in reasoning performance. This also confirms that the necessary reasoning features are already latent within the base model and can be elicited with high efficiency.

Overall, this presents a trade-off: using a lightly RL-trained source yields peak performance, while using the base model enables an efficient, end-to-end workflow that still delivers competitive results.

## 4.2 EXPERIMENTS ON LLAMA-STYLE ARCHITECTURES

To assess the broader applicability of SAE-Tuning, we tested its effectiveness beyond the Qwen model family by applying it to Llama-style architectures. For this experiment, we used the STILL dataset within our SAE-Tuning framework on three variants of the OctoThinker-3B-Base models (Wang et al., 2025c). Each variant serves as the source and target model at the same time.

| MODEL NAME | AMC23 | MATH500 |
|---|---|---|
| OctoThinker-3B-Base-Long | 7.50 | 25.80 |
| Resa-STILL-OctoThinker-3B-Base-Long | 22.50 | 30.00 |
| OctoThinker-3B-Base-Short | 2.50 | 31.40 |
| Resa-STILL-OctoThinker-3B-Base-Short | 27.50 | 38.20 |
| OctoThinker-3B-Base-Hybrid | 10.00 | 30.80 |
| Resa-STILL-OctoThinker-3B-Base-Hybrid | 27.50 | 40.80 |

| MODEL NAME | AMC23 | MATH500 |
|---|---|---|
| OctoThinker-8B-Base-Long | 5.00 | 37.80 |
| Resa-STILL-OctoThinker-8B-Base-Long | 13.50 | 36.70 |
| OctoThinker-8B-Base-Short | 7.50 | 38.60 |
| Resa-STILL-OctoThinker-8B-Base-Short | 12.50 | 40.40 |
| OctoThinker-8B-Base-Hybrid | 5.00 | 42.60 |
| Resa-STILL-OctoThinker-8B-Base-Hybrid | 10.00 | 45.80 |

Table 3: **SAE-Tuning on Llama-Style Models** Performance evaluation of SAE-Tuning on OctoThinker-3B-Base and OctoThinker-8B-Base models, indicating the effectiveness of SAE-Tuning on models besides Qwen. More detailed results are shown in Appendix C.5.

SAE-Tuning proves effective across both 3B and 8B Llama-style architectures in Table 3. For OctoThinker-3B, it delivers large gains on AMC23 (up to +25 points, including a tenfold jump from 2.50% to 27.50% on the Short variant) and consistent improvements on MATH500 (e.g., +10 points on the Hybrid model). For OctoThinker-8B, where baselines are already stronger, SAE-Tuning still yields meaningful increases: +5 to 8.5 points on AMC23 and up to +3.2 points on MATH500 (Hybrid: 42.60% to 45.80%), with only the Long variant showing a small dip on MATH500. These results confirm that SAE-Tuning generalizes beyond the Qwen family and provides reliable reasoning improvements across distinct Llama-style models and scales.

## 5 HYPOTHESIS: GENERALIZABLE AND MODULAR REASONING ABILITY

We now show that the reasoning ability captured by SAE-Tuning is a generalizable and modular skill. We formulate this as a claim: *Reasoning abilities extracted via SAEs can be transferred across both data distributions and models.* To validate this, we conduct two sets of experiments: First, we test out-of-distribution generalization by applying reasoning extracted from one dataset to another. Second, we test cross-model transfer by applying reasoning extracted from one model to another.

**Out-of-Distribution Generalization** To assess out-of-distribution (OOD) generalization, we use a single dataset, STILL, to train the SAE on the source model (the "trigger" step). We then use that trained SAE to guide a SFT process of the target model on a completely different dataset (the "elicit" step). We test this on datasets that have varying degrees of overlap with STILL. Specifically, DeepScaleR fully covers the STILL dataset (which we refer as the coverage dataset) while Open-S1 (Dang & Ngo, 2025), II-Thought (Internet, 2025), and OpenR1 (Hugging Face, 2025) have underlying overlapped sources with STILL (which we coin as the intersection datasets). As shown in Table 4, the Resa-STILL2*X* models, where reasoning ability from STILL is transferred to a new dataset *X*, consistently achieve performance on par with models trained end-to-end via RL on that new dataset. For example, Resa-STILL2DeepScaleR scores 48.77%, almost identical to Tina-DeepScaleR (48.38%) which was trained entirely on DeepScaleR. This pattern holds across

| OUT-OF-DISTRIBUTION COVERAGE DATA | AIME24 | AIME25 | AMC23 | MATH500 | GPQA | MINERVA | AVG. |
|---|---|---|---|---|---|---|---|
| DeepScaleR-1.5B-Preview | 36.67 | 26.67 | 77.50 | 87.80 | 31.82 | 31.99 | 48.74 |
| Tina-DeepScaleR | 43.33 | 26.67 | 67.50 | 86.20 | 37.88 | 28.68 | 48.38 |
| Resa-STILL2DeepScaleR (i.e., Resa-DeepScaleR-v4) | 33.33 | 30.00 | 80.00 | 84.00 | 29.41 | 35.86 | 48.77 |
| OUT-OF-DISTRIBUTION INTERSECTION DATA | AIME24 | AIME25 | AMC23 | MATH500 | GPQA | MINERVA | AVG. |
| Open-RS1 | 26.67 | 20.00 | 72.50 | 83.60 | 28.68 | 35.35 | 44.47 |
| Tina-Open-S1 | 43.33 | 20.00 | 80.00 | 84.00 | 28.68 | 35.35 | 48.56 |
| Resa-STILL2Open-S1 | 36.67 | 23.33 | 85.00 | 84.60 | 30.88 | 31.82 | 48.72 |
| II-Thought-1.5B-Preview | 30.00 | 23.33 | 72.50 | 86.80 | 30.88 | 31.90 | 45.90 |
| Tina-II-Thought | 40.00 | 20.00 | 80.00 | 86.00 | 33.84 | 26.84 | 47.78 |
| Resa-STILL2II-Thought | 40.00 | 23.33 | 75.00 | 83.20 | 31.25 | 38.89 | 48.61 |
| Tina-OpenR1 | 36.67 | 26.67 | 75.00 | 86.80 | 30.51 | 39.90 | 49.26 |
| Resa-STILL2OpenR1 | 33.33 | 30.00 | 77.50 | 86.80 | 27.21 | 41.92 | 49.46 |
| REASONING-AS-AN-ADAPTER | AIME24 | AIME25 | AMC23 | MATH500 | GPQA | MINERVA | AVG. |
| Resa-STILL-Qwen-Math-Adapter | 36.67 | 20.00 | 82.50 | 83.40 | 31.25 | 33.33 | 47.86 |
| Resa-STILL-Qwen-Adapter | 30.00 | 30.00 | 72.50 | 85.60 | 31.25 | 35.86 | 47.54 |

Table 4: **Generality and Modularity of Reasoning Ability** (Top & middle) The results demonstrate OOD generalization across datasets. Resa-STILL2*X* models are trained by extracting reasoning from the STILL dataset and applying it to a new elicitation dataset *X*. (Bottom) The results demonstrate cross-model transfer. A reasoning adapter trained on Qwen-Math or Qwen is transferred to R1-Distill at inference time. More detailed results are shown in Appendix C.6.

all tested datasets. This robust performance demonstrates that the reasoning features extracted from the STILL dataset are not overfit to its specific data distribution. They represent a more general reasoning process that can be effectively applied to new distributions, showcasing OOD resilience.

**Modular Reasoning-as-an-Adapter** Recall from Section 3.1 that during SAE-guided SFT, the parameters we train are all from low-rank adapters. Therefore, we explore if the extracted reasoning ability can similarly be treated as a modular "adapter" that can be plugged into other model. Specifically, we perform SAE-Tuning on models like Qwen-Math (Yang et al., 2024) and Qwen (Qwen et al., 2025) to produce a set of adapters. Then, at test time, we attach such adapters to R1-Distill in the same family, without any further training. The models in this family share an architecture but differ in their foundational knowledge: R1-Distill has the most general knowledge, Qwen-Math is specialized with math data, and Qwen is the most basic. This tests whether our extracted reasoning abilities can be separated from the foundational knowledge of the model it was trained on. As shown in the final rows of Table 4, the adapter trained on Qwen-Math or Qwen and attached to R1-Distill achieves an average score of 47.86% or 47.54%, respectively. This performance is competitive with models where the entire SAE-Tuning process was performed directly on R1-Distill (e.g., Resa-STILL-v1, 47.28% avg). This result provides evidence that:

$$\text{Strong Reasoning Model} \approx \text{Abstract Reasoning Ability} + \text{Foundational Knowledge.}$$

Our SAE-Tuning procedure aims to isolate the "Abstract Reasoning Ability" component into a portable adapter and the final performance is then a direct combination of this adapter with a model that possesses sufficient "Foundational Knowledge." This opens up possibilities for creating highly capable and efficient models by composing reasoning abilities and foundational knowledge.

# 6 CONCLUSION

In this work, we moved beyond the prevailing paradigms of resource-intensive RL and quality-sensitive CoT-based SFT. Specifically, we introduced SAE-Tuning, a novel procedure that leverages SAEs to identify, extract, and elicit latent reasoning abilities using only CoT-free data. Our extensive experiments validated this approach on three key fronts. First, we demonstrated that SAE-Tuning is a performant and practical method, capable of not only replicating the performance of RL-trained models but, more importantly, of eliciting equivalent reasoning abilities directly from certain base models. This process also extends across architectures for Llama-style models. Second, we established the surprising generality of these extracted abilities, demonstrating both their robustness to out-of-distribution data and also their modularity as portable "reasoning adapters."

# 7 REPRODUCIBILITY STATEMENT

All experiments and results presented in this paper are reproducible. We have included our complete source code as supplementary material, which relies solely on publicly available datasets and open-source libraries. The code is structured to allow for easy replication of our experiments and ablation studies. Upon acceptance, we will release the non-anonymized code and pre-trained models under a permissive open-source license in a public repository.

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

# APPENDIX

## A   THE USE OF LARGE LANGUAGE MODELS (LLMS)

We declare that LLMs were used to assist with grammar, phrasing, and polishing of the manuscript's text. All core concepts, methodologies, and scientific contributions are entirely the work of the authors.

## B   ADDITIONAL EXPERIMENTAL DETAILS

### B.1   PRACTICAL IMPLEMENTATION SETUP

| EXPERIMENTAL TASK | TRAINING COST EST. | EVALUATION COST EST. | TOTAL COST EST. |
|---|---|---|---|
| **Main: Resa-STILL-v1 (in Table 7)** | $1 | $4 | $5 |
| **Main: Resa-STILL-v2 (in Table 2)** | $1 | $4 | $5 |
| **Main: Resa-STILL-v3 (in Table 2)** | $1 | $4 | $5 |
| **Main: Resa-STILL-v4 (in Table 2)** | $1 | $4 | $5 |
| **Main: Resa-STILL-v5 (in Table 2)** | $1 | $4 | $5 |
| **Main: Resa-DeepScaleR-v1 (in Table 7)** | $1 | $6 | $7 |
| **Main: Resa-DeepScaleR-v2 (in Table 2)** | $1 | $6 | $7 |
| **Main: Resa-DeepScaleR-v3 (in Table 2)** | $1 | $6 | $7 |
| **Main: Resa-DeepScaleR-v4 (in Table 4)** | $1 | $6 | $7 |
| **Ablation: Algorithm (in Table 7)** | $2 | $10 | $12 |
| **Ablation: Source Model (in Table 2)** | $14 | $60 | $74 |
| **Ablation: SAE Training Mode (in Table 2)** | $3 | $12 | $15 |
| **Hypothesis: Generality (in Table 4)** | $7 | $35 | $42 |
| **Hypothesis: Modularity (in Table 4)** | $102 | $8 | $120 |
| **Hypothesis: Transparency (in Table 8)** | $26 | $104 | $130 |
| **Total: All Tasks** | **$163** | **$273** | **$436** |
| **Total: Main Tasks** | **$9** | **$44** | **$53** |
| **Total: Best Resa Model** | **$1** | **$4** | **$5** |

Table 5: **Computational Cost Breakdown** We provide a detailed cost breakdown of all experiments in this paper. Notice that the training cost estimate includes the costs for training both models and SAEs.

**Overall Budget** A primary motivation for developing SAE-Tuning is to democratize research into reasoning models by establishing a low-cost and high-efficiency paradigm via SAEs. We deliberately constrain our setup to a minimal hardware footprint, using just 2 NVIDIA L40S or NVIDIA RTX 6000 Ada GPUs for all training and evaluation tasks. This setup is readily accessible on major cloud platforms, with an approximate cost of $1 USD per GPU hour at the time of our experiments. As detailed in Table 5, this approach demonstrates high cost-efficiency. We believe this setup provides a valuable testbed for the broader research community.

## B.2 FULL HYPERPARAMETER

We show our default choice of hyperparameters in Table 6. The differences between main and ablation experiments largely lie in the hyperparameters we ablate over, which means that most of following hyperparameters are held constant across all experiments.

| SAE-Tuning Stage I: SAE Training | |
| --- | --- |
| Base model hidden size | 1536 |
| Number of features | 65536 |
| Dead feature threshold | 1e6 |
| Top-$k$ value | 32 |
| Decoder normalization | True |
| Optimizer | Signum |
| Epochs | 1 |
| Batch Size | 16 |
| Learning Rate | 2.5e-4 |
| Learning Rate Scheduler | Constant |
| SAE-Tuning Stage II: SAE-Guided SFT | |
| LoRA Modules | query, key, value, dense |
| LoRA Rank | 32 |
| LoRA $\alpha$ | 128 |
| LoRA Dropout | 0.05 |
| Optimizer | AdamW |
| Optimizer Momentum | $\beta_1, \beta_2 = 0.9, 0.999$ |
| Epochs | 2 |
| Batch Size | 1 |
| Learning Rate | 1e-6 |
| Learning Rate Scheduler | Cosine with Min LR |

Table 6: **Default Hyperparameter Settings of SAE-Tuning** (Top) The default setting of SAE training. (Bottom) The default setting of SAE-Guided SFT.

## C  ADDITIONAL EXPERIMENT RESULTS

By default, for the following experiments, we choose the best Tina checkpoint trained on a specific dataset as the source model. The target model is by default the R1-Distill. For SAE, unless stated otherwise, we follow the "fine-tuned" training mode. The SAE is hooked to the output of the multilayer perceptron submodule after the 12th layer (out of 28) of the source model.

### C.1  REASONING ABILITY REPLICATION

In this experiment, we aim to answer: *Can SAE-Tuning extract and transfer reasoning ability from a source model post-trained for reasoning via RL?* Therefore, we use the Tina models (Wang et al., 2025a), which were trained with RL from R1-Distill on different datasets, as our source models. The goal is to see if our Resa models can match the Tina models' performance.

| MODEL NAME | AIME24 | AIME25 | AMC23 | MATH500 | GPQA | MINERVA | AVG. |
|---|---|---|---|---|---|---|---|
| DeepSeek-R1-Distilled-Qwen-1.5B | 23.33 | 16.67 | 62.50 | 82.60 | 31.82 | 30.15 | 41.18 |
| STILL-3-1.5B-preview | 26.67 | 26.67 | 67.50 | 86.40 | 34.34 | 27.57 | 44.86 |
| Tina-STILL (CoT-free RL) | 36.67 | 30.00 | 77.50 | 84.60 | 33.33 | 26.84 | 48.16 |
| Resa-STILL-v1 (CoT-free SAE-Tuning) | 33.33 | 33.33 | 75.00 | 83.80 | 29.41 | 28.79 | 47.28 |
| DeepScaleR-1.5B-Preview | 36.67 | 26.67 | 77.50 | 87.80 | 31.82 | 31.99 | 48.74 |
| Tina-DeepScaleR (CoT-free RL) | 43.33 | 26.67 | 67.50 | 86.20 | 37.88 | 28.68 | 48.38 |
| Resa-DeepScaleR-v1 (CoT-free SAE-Tuning) | 36.67 | 23.33 | 85.00 | 83.00 | 32.35 | 33.33 | 48.95 |
| ALGORITHM ABLATION | AIME24 | AIME25 | AMC23 | MATH500 | GPQA | MINERVA | AVG. |
| STILL-CoT-free-SFT | 20.00 | 16.67 | 60.00 | 81.20 | 29.78 | 26.36 | 39.00 |
| DeepScaleR-CoT-based-SFT | 10.00 | 6.67 | 57.50 | 68.60 | 20.22 | 36.36 | 33.22 |

Table 7: **Reasoning Ability Replication**  SAE-Tuning successfully replicates the performance of RL-trained source models. Resa models (trained with SAE-Tuning on CoT-free data) achieve performance on par with or exceeding their Tina counterparts. More details are shown in Appendix C.3.

Table 7 summarizes our proof of concept results. On the STILL dataset, our Resa-STILL-v1 (47.28% avg) recovers 98.2% of the performance of the RL-trained Tina-STILL (48.16% avg). On the DeepScaleR dataset, our Resa-DeepScaleR-v1 (48.95% avg) not only replicates but slightly surpasses the performance of its corresponding Tina-DeepScaleR source model (48.38% avg). The algorithm ablation clearly demonstrates the necessity of our method using SAEs: standard SFT on the same CoT-free data (i.e., STILL-CoT-free-SFT) achieves a mere 39.00% average, falling far short of both the Tina models and our Resa counterparts. This shows that simply training on the final answers is insufficient and the SAE-guided SFT is the critical ingredient for eliciting reasoning abilities. Furthermore, standard SFT on a CoT-based dataset (i.e., DeepScaleR-CoT-based-SFT) also performs worse (33.22% avg), suggesting that naive CoT-based training is not an effective strategy for improving reasoning and underscores the novelty of our CoT-free approach via SAE-Tuning.

### C.2  HYPOTHESIS: LAYERWISE REASONING FEATURE EXTRACTION

A claim of SAE-Tuning is that it provides a transparent approach to reasoning. Having demonstrated that it works, we now investigate how it works. We notice that the performance varies not only depending on the source model but also depending on the specific layer chosen of the source model for SAE training. This moves us beyond heuristics for SAE layer selection to a hypothesis: *The suitability of a model layer for reasoning is predictable and correlated with the presence of quantifiable reasoning features.* To test this hypothesis, we introduce a novel prompt-only reasoning feature extraction method and use it to establish the underlying correlation between these features and reasoning performance.

**Prompt-Only Reasoning Feature Extraction**  We propose a novel method to explicitly identify and quantify "reasoning features" and test if their distribution predicts the final performance of a Resa model. We hypothesize that features specifically involved in reasoning should activate primarily when the model is prompted to "think." Specifically, we pass the standard DeepSeek-R1 system prompt containing <think> and </think> tokens through a model equipped with trained SAEs inserted after each layer indexed from 2 to 27. We cut off the first and final layer from the total 28

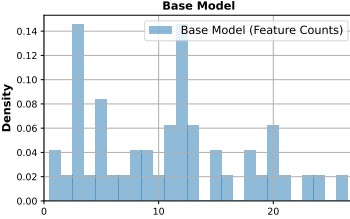 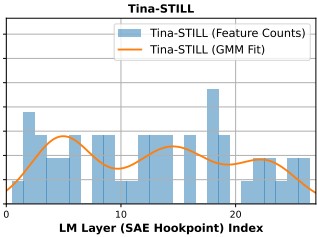 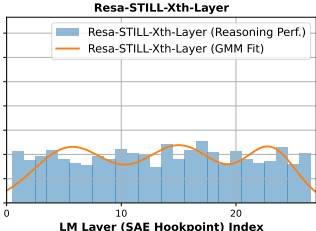

Figure 3: **Reasoning Feature Extraction** (Left) This shows the layer-wise feature counts of the base R1-Distill model. (Middle) This shows the layer-wise feature counts of the Tina-STILL model. (Right) This shows the reasoning performance of the trained Resa models with different layer-wise SAEs when Tina-STILL is the source model.

layers since these two layers are mainly used for embedding and next token prediction, respectively. We then define reasoning features as those SAE features that are exclusively and simultaneously activated at the <think> and </think> tokens and not by other parts of the prompt. Applying this method to the base R1-Distill model revealed an interesting pattern that the layer-wise count of these reasoning features exhibits a *tri-modal distribution* around layer indices 3, 12, and 20 as shown in Figure 3.

| MODEL NAME | AIME24 | AIME25 | AMC23 | MATH500 | GPQA | MINERVA | AVG. | Feature Cnts. |
|---|---|---|---|---|---|---|---|---|
| **Resa-STILL-2nd-Layer** | 26.67 | 30.00 | 80.00 | 83.20 | 29.78 | 37.37 | 47.84 | 1 |
| **Resa-STILL-3rd-Layer** | 26.67 | 36.67 | 70.00 | 83.40 | 28.31 | 33.84 | 46.48 | 4 |
| **Resa-STILL-4th-Layer** | 33.33 | 20.00 | 80.00 | 83.80 | 28.68 | 36.87 | 47.11 | 3 |
| **Resa-STILL-5th-Layer** | 40.00 | 23.33 | 70.00 | 83.20 | 26.84 | 44.95 | 48.05 | 2 |
| **Resa-STILL-6th-Layer** | 33.33 | 23.33 | 72.50 | 83.40 | 31.62 | 35.35 | 46.59 | 2 |
| **Resa-STILL-7th-Layer** | 26.67 | 26.67 | 77.50 | 81.60 | 27.94 | 35.86 | 46.04 | 3 |
| **Resa-STILL-8th-Layer** | 20.00 | 23.33 | 82.50 | 85.20 | 29.78 | 33.33 | 45.69 | 0 |
| **Resa-STILL-9th-Layer** | 36.67 | 20.00 | 80.00 | 84.20 | 27.57 | 34.34 | 47.13 | 3 |
| **Resa-STILL-10th-Layer** | 36.67 | 23.33 | 67.50 | 84.80 | 29.78 | 37.37 | 46.58 | 3 |
| **Resa-STILL-11th-Layer** | 26.67 | 36.67 | 77.50 | 83.80 | 31.25 | 32.83 | 48.12 | 1 |
| **Resa-STILL-12th-Layer** | 30.00 | 26.67 | 77.50 | 84.60 | 31.99 | 34.85 | 47.60 | 1 |
| **Resa-STILL-13th-Layer** | 33.33 | 33.33 | 75.00 | 83.80 | 29.41 | 28.79 | 47.28 | 3 |
| **Resa-STILL-14th-Layer** | 33.33 | 20.00 | 77.50 | 83.60 | 28.68 | 29.80 | 45.48 | 3 |
| **Resa-STILL-15th-Layer** | 36.67 | 23.33 | 80.00 | 84.80 | 30.15 | 38.89 | 48.97 | 3 |
| **Resa-STILL-16th-Layer** | 30.00 | 20.00 | 80.00 | 85.00 | 30.15 | 34.85 | 46.67 | 0 |
| **Resa-STILL-17th-Layer** | 43.33 | 16.67 | 77.50 | 83.80 | 30.88 | 36.36 | 48.09 | 3 |
| **Resa-STILL-18th-Layer** | 43.33 | 20.00 | 77.50 | 84.00 | 33.82 | 37.88 | 49.42 | 0 |
| **Resa-STILL-19th-Layer** | 33.33 | 30.00 | 72.50 | 84.20 | 29.04 | 36.87 | 47.66 | 5 |
| **Resa-STILL-20th-Layer** | 33.33 | 23.33 | 75.00 | 85.00 | 29.04 | 29.29 | 45.83 | 3 |
| **Resa-STILL-21st-Layer** | 30.00 | 33.33 | 75.00 | 84.60 | 27.57 | 36.87 | 47.90 | 0 |
| **Resa-STILL-22nd-Layer** | 33.33 | 23.33 | 75.00 | 82.20 | 31.99 | 34.34 | 46.70 | 1 |
| **Resa-STILL-23rd-Layer** | 30.00 | 20.00 | 80.00 | 82.00 | 29.04 | 35.35 | 46.07 | 2 |
| **Resa-STILL-24th-Layer** | 36.67 | 20.00 | 77.50 | 83.80 | 30.51 | 29.80 | 46.38 | 2 |
| **Resa-STILL-25th-Layer** | 40.00 | 30.00 | 70.00 | 84.20 | 28.68 | 37.88 | 48.46 | 1 |
| **Resa-STILL-26th-Layer** | 36.67 | 20.00 | 65.00 | 85.60 | 30.88 | 36.87 | 45.84 | 2 |
| **Resa-STILL-27th-Layer** | 30.00 | 36.67 | 67.50 | 83.00 | 27.94 | 40.40 | 47.59 | 2 |

Table 8: **SAE Hookpoints Ablation** Performance evaluation of Resa-STILL models where SAE-Tuning is applied to each layer indexed from 2 to 27 individually. Feature Cnts. is the number of identified reasoning features in the corresponding layer of the Tina-STILL model. More detailed results are shown in Appendix C.7.

**Feature Counts v.s. Reasoning Performance Correlation** To test the hypothesis that this feature count distribution can predict reasoning performance, we conducted a large-scale study that we created 26 different Resa-STILL models, with each one generated by applying SAE-Tuning to a

different layer of Tina-STILL, from layer 2 to 27. The results in Table 8 confirm that the choice of SAE hookpoint is critical. The average reasoning score fluctuates significantly, ranging from a low of 45.48% (Layer 14) to a high of 49.42% (Layer 18). Also, a naive interpretation, assuming more reasoning features equals better performance, is proven false. For instance, Layer 18 yields the top performance (49.42%) but has 0 identified reasoning features, while Layer 19 has the most features (5) but achieves a lower score (47.66%). This validates our earlier finding that the source of reasoning features is nuanced.

Such result indicates a more complex relationship between reasoning features and reasoning abilities exists. The key insight to discover such relationship comes from analyzing the overall distributions rather than single points. Just as the feature counts across layers form a tri-modal distribution, so does the final reasoning performance. Therefore, we fit a 3-component Gaussian Mixture Model (3-GMM) to both distributions: (1) the a priori reasoning feature counts from the base model, and (2) the final reasoning scores from our 26 Resa models. The GMM analysis reveals an interesting and close structural alignment between the two GMM distributions. The means of the three Gaussian components for the feature count distribution are located near layers 4.9, 14.5, and 22.7. The reasoning performance distribution's components cluster around nearly identical means at layers 5.6, 15.1, and 23.0. This similarity extends to the component weights, which represent the proportion of layers belonging to each cluster. The feature distribution's weights (41%, 37%, 22%) are closely mirrored by the reasoning performance distribution's weights (39%, 37%, 24%). The overall spread of the two distributions, as measured by entropy, are nearly identical (3.194 for feature counts vs. 3.202 for reasoning performance). This suggests that while a single layer's feature count may not be a good predictor, the overall structure of how reasoning is organized into three distinct layer-clusters within the model is a robust predictor of how performance will be distributed. In practice, one can therefore analyze the source model's feature distribution to strategically identify layer-clusters likely to yield high-performing models, providing a data-driven and transparent method for optimizing the SAE-Tuning process.

## C.3 FULL RESULTS OF TABLE 7

In the following tables, we present the full performance evaluation results of models in Table 7.

| CHECKPOINT STEPS | AIME24 | AIME25 | AMC23 | MATH500 | GPQA | MINERVA | AVG. |
|---|---|---|---|---|---|---|---|
| 1000 | 20.00 | 33.33 | 75.00 | 82.60 | 29.78 | 33.33 | 45.67 |
| 1500 | 33.33 | 23.33 | 75.00 | 82.80 | 30.88 | 30.81 | 46.03 |
| 2000 | 33.33 | 33.33 | 75.00 | 83.80 | 29.41 | 28.79 | 47.28 |
| 2500 | 30.00 | 23.33 | 77.50 | 84.20 | 26.47 | 33.33 | 45.81 |

Table 9: **Performance of Resa-STILL-v1** Each epoch contains 1448 Steps.

| CHECKPOINT STEPS | AIME24 | AIME25 | AMC23 | MATH500 | GPQA | MINERVA | AVG. |
|---|---|---|---|---|---|---|---|
| 500 | 30.00 | 30.00 | 67.50 | 84.00 | 28.68 | 32.32 | 45.42 |
| 1000 | 16.67 | 10.00 | 67.50 | 83.00 | 30.15 | 37.88 | 40.87 |
| 1500 | 23.33 | 20.00 | 70.00 | 84.40 | 27.57 | 37.88 | 43.86 |
| 2000 | 33.33 | 23.33 | 70.00 | 86.00 | 29.04 | 36.87 | 46.43 |
| 2500 | 36.67 | 23.33 | 85.00 | 83.00 | 32.35 | 33.33 | 48.95 |
| 3000 | 20.00 | 26.67 | 65.00 | 83.80 | 30.51 | 35.86 | 43.64 |
| 3500 | 23.33 | 23.33 | 70.00 | 81.80 | 30.15 | 36.36 | 44.16 |
| 4000 | 33.33 | 13.33 | 72.50 | 83.80 | 29.41 | 36.36 | 44.79 |
| 4500 | 36.67 | 13.33 | 70.00 | 83.80 | 27.94 | 31.31 | 43.84 |
| 5000 | 30.00 | 23.33 | 67.50 | 84.40 | 28.68 | 32.32 | 44.37 |

Table 10: **Performance of Resa-DeepScaleR-v1** Each epoch contains 2630 Steps.

| CHECKPOINT STEPS | AIME24 | AIME25 | AMC23 | MATH500 | GPQA | MINERVA | AVG. |
|---|---|---|---|---|---|---|---|
| 500 | 20.00 | 16.67 | 60.00 | 81.20 | 29.78 | 26.36 | 39.00 |
| 1000 | 23.33 | 16.67 | 62.50 | 77.20 | 26.47 | 26.26 | 38.74 |
| 1500 | 13.33 | 10.00 | 60.00 | 74.00 | 28.68 | 27.78 | 35.63 |

Table 11: **Performance of STILL-CoT-free-SFT** Each epoch contains 936 Steps.

| CHECKPOINT STEPS | AIME24 | AIME25 | AMC23 | MATH500 | GPQA | MINERVA | AVG. |
|---|---|---|---|---|---|---|---|
| 1000 | 10.00 | 6.67 | 57.50 | 68.60 | 20.22 | 36.36 | 33.22 |
| 2000 | 16.67 | 6.67 | 52.50 | 67.40 | 21.69 | 28.28 | 32.20 |
| 3000 | 10.00 | 6.67 | 37.50 | 64.20 | 25.37 | 32.32 | 29.34 |
| 4000 | 10.00 | 6.67 | 35.00 | 61.80 | 22.79 | 27.78 | 27.34 |
| 5000 | 10.00 | 0.00 | 32.50 | 64.40 | 23.53 | 29.29 | 26.62 |
| 6000 | 0.00 | 6.67 | 40.00 | 64.00 | 23.53 | 28.79 | 27.16 |
| 7000 | 10.00 | 0.00 | 42.50 | 60.20 | 20.22 | 25.76 | 26.45 |

Table 12: **Performance of DeepScaleR-CoT-based-SFT** Each epoch contains 2520 Steps.

## C.4 Full Results of Table 2

In the following tables, we present the full performance evaluation results of models in Table 2.

| Checkpoint Steps | AIME24 | AIME25 | AMC23 | MATH500 | GPQA | Minerva | Avg. |
|---|---|---|---|---|---|---|---|
| 1000 | 23.33 | 23.33 | 72.50 | 85.40 | 30.51 | 34.85 | 44.99 |
| 1500 | 20.00 | 20.00 | 75.00 | 81.40 | 29.78 | 30.30 | 42.75 |
| 2000 | 23.33 | 23.33 | 67.50 | 83.60 | 29.78 | 35.86 | 43.90 |
| 2500 | 23.33 | 26.67 | 67.50 | 83.00 | 25.74 | 34.34 | 43.43 |

Table 13: **Performance of Resa-STILL-Pretrained-SAE** Each epoch contains 1448 Steps.

| Checkpoint Steps | AIME24 | AIME25 | AMC23 | MATH500 | GPQA | Minerva | Avg. |
|---|---|---|---|---|---|---|---|
| 1000 | 30.00 | 20.00 | 67.50 | 84.40 | 28.31 | 34.85 | 44.18 |
| 1500 | 33.33 | 33.33 | 70.00 | 83.00 | 30.15 | 34.34 | 47.36 |
| 2000 | 33.33 | 26.67 | 72.50 | 81.60 | 30.51 | 29.29 | 45.65 |
| 2500 | 30.00 | 23.33 | 70.00 | 85.60 | 32.35 | 33.33 | 45.77 |

Table 14: **Performance of Resa-STILL-Trained-from-Scratch-SAE** Each epoch contains 1448 Steps.

| Checkpoint Steps | AIME24 | AIME25 | AMC23 | MATH500 | GPQA | Minerva | Avg. |
|---|---|---|---|---|---|---|---|
| 1000 | 23.33 | 23.33 | 77.50 | 82.20 | 28.31 | 31.82 | 44.42 |
| 1500 | 30.00 | 16.67 | 77.50 | 80.40 | 31.25 | 29.80 | 44.27 |
| 2000 | 33.33 | 16.67 | 65.00 | 83.40 | 27.21 | 34.85 | 43.41 |
| 2500 | 23.33 | 20.00 | 77.50 | 84.60 | 27.57 | 35.35 | 44.73 |

Table 15: **Performance of Resa-STILL-Tina-0-step (Fine-tuned SAE)** Each epoch contains 1448 Steps.

| Checkpoint Steps | AIME24 | AIME25 | AMC23 | MATH500 | GPQA | Minerva | Avg. |
|---|---|---|---|---|---|---|---|
| 1000 | 26.67 | 20.00 | 67.50 | 84.00 | 29.04 | 35.35 | 43.76 |
| 1500 | 40.00 | 26.67 | 70.00 | 83.20 | 31.62 | 36.36 | 47.98 |
| 2000 | 26.67 | 26.67 | 65.00 | 83.40 | 27.94 | 36.36 | 44.34 |
| 2500 | 36.67 | 23.33 | 65.00 | 84.00 | 30.88 | 36.36 | 46.04 |

Table 16: **Performance of Resa-STILL-Tina-1-step (Fine-tuned SAE)** Each epoch contains 1448 Steps.

| Checkpoint Steps | AIME24 | AIME25 | AMC23 | MATH500 | GPQA | Minerva | Avg. |
|---|---|---|---|---|---|---|---|
| 1000 | 30.00 | 16.67 | 67.50 | 83.80 | 31.62 | 32.32 | 43.65 |
| 1500 | 23.33 | 23.33 | 75.00 | 82.20 | 29.41 | 33.33 | 44.43 |
| 2000 | 23.33 | 23.33 | 62.50 | 84.20 | 27.57 | 34.85 | 42.63 |
| 2500 | 20.00 | 20.00 | 70.00 | 84.20 | 27.94 | 35.86 | 43.00 |

Table 17: **Performance of Resa-STILL-Tina-10-step (Fine-tuned SAE)** Each epoch contains 1448 Steps.

| Checkpoint Steps | AIME24 | AIME25 | AMC23 | MATH500 | GPQA | Minerva | Avg. |
|---|---|---|---|---|---|---|---|
| 1000 | 26.67 | 23.33 | 70.00 | 83.00 | 28.68 | 38.89 | 45.09 |
| 1500 | 20.00 | 26.67 | 65.00 | 84.60 | 30.51 | 34.85 | 43.60 |
| 2000 | 30.00 | 33.33 | 67.50 | 82.80 | 30.15 | 36.36 | 46.69 |
| 2500 | 33.33 | 26.67 | 72.50 | 82.60 | 27.57 | 38.89 | 46.93 |

Table 18: **Performance of Resa-STILL-Tina-50-step (Fine-tuned SAE)** Each epoch contains 1448 Steps.

| Checkpoint Steps | AIME24 | AIME25 | AMC23 | MATH500 | GPQA | Minerva | Avg. |
|---|---|---|---|---|---|---|---|
| 1000 | 43.33 | 23.33 | 82.50 | 85.60 | 28.68 | 33.33 | 49.46 |
| 1500 | 40.00 | 23.33 | 72.50 | 84.60 | 25.74 | 32.32 | 46.42 |
| 2000 | 30.00 | 16.67 | 70.00 | 85.40 | 33.09 | 33.33 | 44.75 |
| 2500 | 23.33 | 23.33 | 72.50 | 84.40 | 30.15 | 36.36 | 45.01 |

Table 19: **Performance of Resa-STILL-Tina-100-step (Fine-tuned SAE)** Each epoch contains 1448 Steps.

| Checkpoint Steps | AIME24 | AIME25 | AMC23 | MATH500 | GPQA | Minerva | Avg. |
|---|---|---|---|---|---|---|---|
| 1000 | 36.67 | 26.67 | 80.00 | 83.80 | 31.62 | 31.82 | 48.43 |
| 1500 | 30.00 | 23.33 | 82.50 | 84.20 | 31.62 | 36.36 | 48.00 |
| 2000 | 36.67 | 26.67 | 70.00 | 85.60 | 31.99 | 38.89 | 48.30 |
| 2500 | 26.67 | 20.00 | 72.50 | 82.20 | 28.31 | 35.86 | 44.26 |

Table 20: **Performance of Resa-STILL-Tina-500-step (Fine-tuned SAE)** Each epoch contains 1448 Steps.

| Checkpoint Steps | AIME24 | AIME25 | AMC23 | MATH500 | GPQA | Minerva | Avg. |
|---|---|---|---|---|---|---|---|
| 1000 | 30.00 | 20.00 | 75.00 | 83.00 | 26.84 | 32.32 | 44.53 |
| 1500 | 26.67 | 23.33 | 72.50 | 85.40 | 27.94 | 34.85 | 45.11 |
| 2000 | 26.67 | 26.67 | 72.50 | 83.60 | 27.57 | 30.81 | 44.64 |
| 2500 | 23.33 | 26.67 | 67.50 | 82.60 | 30.51 | 36.36 | 44.49 |

Table 21: **Performance of Resa-STILL-Tina-3000-step (Fine-tuned SAE)** Each epoch contains 1448 Steps.

| CHECKPOINT STEPS | AIME24 | AIME25 | AMC23 | MATH500 | GPQA | MINERVA | AVG. |
|---|---|---|---|---|---|---|---|
| 1000 | 33.33 | 26.67 | 70.00 | 87.00 | 29.41 | 41.92 | 48.06 |
| 1500 | 23.33 | 23.33 | 70.00 | 84.00 | 29.78 | 34.85 | 44.22 |
| 2000 | 36.67 | 23.33 | 72.50 | 83.40 | 28.68 | 36.87 | 46.91 |
| 2500 | 43.33 | 23.33 | 65.00 | 84.20 | 27.94 | 29.80 | 45.60 |

Table 22: **Performance of Resa-STILL-Tina-0-step (Trained-from-Scratch SAE)** Each epoch contains 1448 Steps.

| CHECKPOINT STEPS | AIME24 | AIME25 | AMC23 | MATH500 | GPQA | MINERVA | AVG. |
|---|---|---|---|---|---|---|---|
| 1000 | 30.00 | 23.33 | 75.00 | 83.80 | 25.74 | 34.85 | 45.45 |
| 1500 | 26.67 | 23.33 | 72.50 | 85.00 | 29.78 | 38.38 | 45.94 |
| 2000 | 33.33 | 23.33 | 65.00 | 82.00 | 29.78 | 34.85 | 44.72 |
| 2500 | 33.33 | 33.33 | 72.50 | 82.20 | 29.41 | 35.35 | 47.69 |

Table 23: **Performance of Resa-STILL-Tina-1-step (Trained-from-Scratch SAE)** Each epoch contains 1448 Steps.

| CHECKPOINT STEPS | AIME24 | AIME25 | AMC23 | MATH500 | GPQA | MINERVA | AVG. |
|---|---|---|---|---|---|---|---|
| 1000 | 26.67 | 20.00 | 67.50 | 85.80 | 28.31 | 36.87 | 44.19 |
| 1500 | 33.33 | 16.67 | 67.50 | 86.20 | 30.51 | 37.37 | 45.26 |
| 2000 | 33.33 | 23.33 | 67.50 | 84.40 | 29.41 | 32.32 | 45.05 |
| 2500 | 26.67 | 30.00 | 62.50 | 82.20 | 26.84 | 34.85 | 43.84 |

Table 24: **Performance of Resa-STILL-Tina-10-step (Trained-from-Scratch SAE)** Each epoch contains 1448 Steps.

| CHECKPOINT STEPS | AIME24 | AIME25 | AMC23 | MATH500 | GPQA | MINERVA | AVG. |
|---|---|---|---|---|---|---|---|
| 1000 | 36.67 | 23.33 | 80.00 | 84.20 | 28.68 | 35.86 | 48.12 |
| 1500 | 23.33 | 33.33 | 72.50 | 84.00 | 27.21 | 35.86 | 46.04 |
| 2000 | 30.00 | 30.00 | 70.00 | 83.20 | 32.72 | 37.37 | 47.22 |
| 2500 | 43.33 | 23.33 | 77.50 | 83.40 | 29.41 | 38.89 | 49.31 |

Table 25: **Performance of Resa-STILL-Tina-50-step (Trained-from-Scratch SAE)** Each epoch contains 1448 Steps.

| CHECKPOINT STEPS | AIME24 | AIME25 | AMC23 | MATH500 | GPQA | MINERVA | AVG. |
|---|---|---|---|---|---|---|---|
| 1000 | 33.33 | 23.33 | 90.00 | 82.60 | 28.68 | 35.35 | 48.88 |
| 1500 | 46.67 | 20.00 | 62.50 | 83.00 | 28.31 | 31.31 | 45.30 |
| 2000 | 36.67 | 20.00 | 75.00 | 83.20 | 30.51 | 38.38 | 47.29 |
| 2500 | 30.00 | 23.33 | 65.00 | 83.20 | 29.04 | 35.35 | 44.32 |

Table 26: **Performance of Resa-STILL-Tina-100-step (Trained-from-Scratch SAE)** Each epoch contains 1448 Steps.

| CHECKPOINT STEPS | AIME24 | AIME25 | AMC23 | MATH500 | GPQA | MINERVA | AVG. |
|---|---|---|---|---|---|---|---|
| 1000 | 23.33 | 20.00 | 75.00 | 84.60 | 30.51 | 33.84 | 44.55 |
| 1500 | 30.00 | 20.00 | 70.00 | 83.80 | 30.51 | 32.32 | 44.44 |
| 2000 | 36.67 | 20.00 | 67.50 | 84.20 | 30.88 | 35.35 | 45.77 |
| 2500 | 20.00 | 23.33 | 67.50 | 82.80 | 28.68 | 35.35 | 42.94 |

Table 27: **Performance of Resa-STILL-Tina-500-step (Trained-from-Scratch SAE)** Each epoch contains 1448 Steps.

| CHECKPOINT STEPS | AIME24 | AIME25 | AMC23 | MATH500 | GPQA | MINERVA | AVG. |
|---|---|---|---|---|---|---|---|
| 1000 | 16.67 | 23.33 | 65.00 | 83.60 | 30.88 | 34.85 | 42.39 |
| 1500 | 30.00 | 20.00 | 77.50 | 86.20 | 31.62 | 37.88 | 47.20 |
| 2000 | 33.33 | 26.67 | 65.00 | 85.20 | 31.62 | 35.35 | 46.20 |
| 2500 | 20.00 | 26.67 | 67.50 | 82.40 | 28.68 | 32.83 | 43.01 |

Table 28: **Performance of Resa-STILL-Tina-3000-step (Trained-from-Scratch SAE)** Each epoch contains 1448 Steps.

| CHECKPOINT STEPS | AIME24 | AIME25 | AMC23 | MATH500 | GPQA | MINERVA | AVG. |
|---|---|---|---|---|---|---|---|
| 1000 | 30.00 | 23.33 | 70.00 | 86.40 | 27.94 | 32.83 | 45.08 |
| 1500 | 33.33 | 23.33 | 80.00 | 86.00 | 30.51 | 31.31 | 47.41 |
| 2000 | 30.00 | 16.67 | 77.50 | 83.60 | 28.31 | 31.82 | 44.65 |
| 2500 | 23.33 | 20.00 | 75.00 | 82.00 | 29.78 | 36.36 | 44.41 |
| 3000 | 26.67 | 16.67 | 72.50 | 83.20 | 31.62 | 33.33 | 44.00 |
| 3500 | 30.00 | 23.33 | 75.00 | 85.80 | 27.94 | 30.80 | 45.48 |

Table 29: **Performance of Resa-DeepScaleR-Best-Tina-1000-step (Trained-from-Scratch SAE)** Each epoch contains 1914 Steps.

| CHECKPOINT STEPS | AIME24 | AIME25 | AMC23 | MATH500 | GPQA | MINERVA | AVG. |
|---|---|---|---|---|---|---|---|
| 1000 | 26.67 | 16.67 | 75.00 | 84.60 | 28.68 | 31.80 | 43.90 |
| 1500 | 36.67 | 30.00 | 77.50 | 83.80 | 29.41 | 31.26 | 48.11 |
| 2000 | 23.33 | 20.00 | 82.50 | 82.40 | 28.68 | 34.36 | 45.21 |
| 2500 | 33.33 | 23.33 | 67.50 | 83.00 | 29.04 | 32.43 | 44.77 |
| 3000 | 40.00 | 20.00 | 72.50 | 84.40 | 26.47 | 35.35 | 46.45 |
| 3500 | 40.00 | 30.00 | 75.00 | 84.00 | 30.15 | 33.33 | 48.75 |

Table 30: **Performance of Resa-DeepScaleR-Tina-0-step (Trained-from-Scratch SAE)** Each epoch contains 1914 Steps.

## C.5 Full Results of Table 3

| Checkpoint Steps | AMC23 | MATH500 |
|---|---|---|
| 500 | 10.00 | 30.20 |
| 1000 | 12.50 | 30.00 |
| 1500 | 10.00 | 28.60 |
| 2000 | 22.50 | 30.00 |
| 2500 | 12.50 | 30.20 |

Table 31: **Performance of Resa-STILL-OctoThinker-3B-Base-Long** Each epoch contains 1063 Steps.

| Checkpoint Steps | AMC23 | MATH500 |
|---|---|---|
| 500 | 27.50 | 38.20 |
| 1000 | 20.00 | 37.60 |
| 1500 | 17.50 | 38.20 |
| 2000 | 10.00 | 38.20 |
| 2500 | 22.50 | 35.80 |

Table 32: **Performance of Resa-STILL-OctoThinker-3B-Base-Short** Each epoch contains 1063 Steps.

| Checkpoint Steps | AMC23 | MATH500 |
|---|---|---|
| 500 | 20.00 | 34.80 |
| 1000 | 27.50 | 40.80 |
| 1500 | 20.00 | 39.20 |
| 2000 | 27.50 | 37.40 |
| 2500 | 17.50 | 42.00 |

Table 33: **Performance of Resa-STILL-OctoThinker-3B-Base-Hybrid** Each epoch contains 1063 Steps.

## C.6 Full Results of Table 4

In the following tables, we present the full performance evaluation results of models in Table 4.

| Checkpoint Steps | AIME24 | AIME25 | AMC23 | MATH500 | GPQA | Minerva | Avg. |
|---|---|---|---|---|---|---|---|
| 500 | 33.33 | 30.00 | 80.00 | 84.00 | 29.41 | 35.86 | 48.77 |
| 1000 | 33.33 | 23.33 | 70.00 | 84.80 | 29.41 | 32.83 | 45.62 |
| 1500 | 26.67 | 13.33 | 67.50 | 83.80 | 31.25 | 36.87 | 43.24 |
| 2000 | 26.67 | 23.33 | 70.00 | 83.20 | 28.68 | 34.34 | 44.37 |
| 2500 | 33.33 | 16.67 | 70.00 | 82.40 | 27.57 | 35.35 | 44.22 |
| 3000 | 30.00 | 26.67 | 57.50 | 83.20 | 28.68 | 37.37 | 43.90 |
| 3500 | 26.67 | 13.33 | 77.50 | 83.20 | 28.31 | 34.85 | 43.98 |
| 4000 | 30.00 | 23.33 | 60.00 | 84.40 | 26.84 | 37.37 | 43.66 |
| 4500 | 36.67 | 20.00 | 75.00 | 83.80 | 27.94 | 37.37 | 46.80 |
| 5000 | 36.67 | 20.00 | 72.50 | 84.00 | 26.84 | 31.82 | 45.31 |

Table 34: **Performance of Resa-STILL2DeepScaleR**  Each epoch contains 1914 Steps.

| Checkpoint Steps | AIME24 | AIME25 | AMC23 | MATH500 | GPQA | Minerva | Avg. |
|---|---|---|---|---|---|---|---|
| 500 | 23.33 | 20.00 | 72.50 | 84.00 | 30.51 | 32.83 | 43.86 |
| 1000 | 36.67 | 23.33 | 85.00 | 84.60 | 30.88 | 31.82 | 48.72 |
| 1500 | 33.33 | 26.67 | 72.50 | 83.40 | 29.41 | 39.90 | 47.54 |
| 2000 | 30.00 | 26.67 | 75.00 | 84.00 | 30.88 | 37.88 | 47.41 |

Table 35: **Performance of Resa-STILL2Open-S1**  Each epoch contains 1063 Steps.

| Checkpoint Steps | AIME24 | AIME25 | AMC23 | MATH500 | GPQA | Minerva | Avg. |
|---|---|---|---|---|---|---|---|
| 1000 | 30.00 | 20.00 | 75.00 | 84.60 | 27.21 | 34.34 | 45.19 |
| 2000 | 40.00 | 23.33 | 75.00 | 83.20 | 31.25 | 38.89 | 48.61 |
| 3000 | 26.67 | 23.33 | 75.00 | 85.20 | 28.31 | 36.36 | 45.81 |
| 4000 | 26.67 | 13.33 | 67.50 | 85.80 | 27.94 | 41.41 | 43.78 |
| 5000 | 26.67 | 20.00 | 72.50 | 85.60 | 29.41 | 37.37 | 45.26 |

Table 36: **Performance of Resa-STILL2II-Thought**  Each epoch contains 2664 Steps.

| CHECKPOINT STEPS | AIME24 | AIME25 | AMC23 | MATH500 | GPQA | MINERVA | AVG. |
|---|---|---|---|---|---|---|---|
| 1000 | 33.33 | 16.67 | 72.50 | 84.00 | 31.99 | 40.91 | 46.57 |
| 2000 | 33.33 | 30.00 | 77.50 | 86.80 | 27.21 | 41.92 | 49.46 |
| 3000 | 30.00 | 30.00 | 72.50 | 84.60 | 29.04 | 33.84 | 46.66 |
| 4000 | 33.33 | 23.33 | 65.00 | 84.20 | 29.04 | 34.34 | 44.88 |
| 5000 | 33.33 | 23.33 | 67.50 | 84.40 | 28.68 | 32.32 | 44.93 |
| 6000 | 20.00 | 20.00 | 72.50 | 84.20 | 28.68 | 31.31 | 42.78 |
| 7000 | 23.33 | 20.00 | 67.50 | 81.40 | 30.88 | 37.88 | 43.50 |
| 8000 | 33.33 | 23.33 | 72.50 | 80.40 | 26.10 | 35.86 | 45.25 |
| 9000 | 30.00 | 23.33 | 70.00 | 83.60 | 27.94 | 30.81 | 44.28 |

Table 37: **Performance of Resa-STILL2OpenR1** Each epoch contains 4911 Steps.

| CHECKPOINT STEPS | AIME24 | AIME25 | AMC23 | MATH500 | GPQA | MINERVA | AVG. |
|---|---|---|---|---|---|---|---|
| 1000 | 36.67 | 20.00 | 82.50 | 83.40 | 31.25 | 33.33 | 47.86 |
| 1500 | 36.67 | 16.67 | 67.50 | 86.20 | 31.25 | 34.85 | 45.52 |
| 2000 | 40.00 | 20.00 | 72.50 | 84.60 | 30.15 | 32.32 | 46.60 |
| 2500 | 26.67 | 16.67 | 72.50 | 84.60 | 26.84 | 35.86 | 43.86 |

Table 38: **Performance of Resa-STILL-Qwen-Math-Adapter** Each epoch contains 1448 Steps.

| CHECKPOINT STEPS | AIME24 | AIME25 | AMC23 | MATH500 | GPQA | MINERVA | AVG. |
|---|---|---|---|---|---|---|---|
| 1000 | 30.00 | 30.00 | 72.50 | 85.60 | 31.25 | 35.86 | 47.54 |
| 1500 | 20.00 | 20.00 | 72.50 | 83.00 | 30.15 | 35.35 | 43.50 |
| 2000 | 26.67 | 30.00 | 67.50 | 84.60 | 25.74 | 32.83 | 44.56 |
| 2500 | 30.00 | 16.67 | 70.00 | 83.60 | 29.78 | 34.34 | 44.07 |

Table 39: **Performance of Resa-STILL-Qwen-Adapter** Each epoch contains 1448 Steps.

## C.7 FULL RESULTS OF TABLE 8

In the following tables, we present the full performance evaluation results of models in Table 8.

| CHECKPOINT STEPS | AIME24 | AIME25 | AMC23 | MATH500 | GPQA | MINERVA | AVG. |
|---|---|---|---|---|---|---|---|
| 1000 | 26.67 | 20.00 | 70.00 | 85.00 | 29.41 | 32.32 | 43.90 |
| 1500 | 26.67 | 16.67 | 70.00 | 82.40 | 26.84 | 31.31 | 42.31 |
| 2000 | 26.67 | 30.00 | 80.00 | 83.20 | 29.78 | 37.37 | 47.84 |
| 2500 | 16.67 | 23.33 | 67.50 | 86.00 | 32.35 | 34.34 | 43.37 |

Table 40: **Performance of Resa-STILL-2nd-Layer** Each epoch contains 1448 Steps.

| CHECKPOINT STEPS | AIME24 | AIME25 | AMC23 | MATH500 | GPQA | MINERVA | AVG. |
|---|---|---|---|---|---|---|---|
| 1000 | 20.00 | 20.00 | 82.50 | 84.60 | 24.26 | 32.32 | 43.95 |
| 1500 | 26.67 | 36.67 | 70.00 | 83.40 | 28.31 | 33.84 | 46.48 |
| 2000 | 33.33 | 23.33 | 70.00 | 83.40 | 28.68 | 33.84 | 45.43 |
| 2500 | 26.67 | 16.67 | 72.50 | 84.60 | 31.25 | 30.30 | 43.66 |

Table 41: **Performance of Resa-STILL-3rd-Layer** Each epoch contains 1448 Steps.

| CHECKPOINT STEPS | AIME24 | AIME25 | AMC23 | MATH500 | GPQA | MINERVA | AVG. |
|---|---|---|---|---|---|---|---|
| 1000 | 26.67 | 16.67 | 67.50 | 84.20 | 28.31 | 40.91 | 44.04 |
| 1500 | 33.33 | 20.00 | 80.00 | 83.80 | 28.68 | 36.87 | 47.11 |
| 2000 | 36.67 | 23.33 | 70.00 | 83.20 | 27.57 | 34.85 | 45.94 |
| 2500 | 40.00 | 20.00 | 72.50 | 83.80 | 30.88 | 31.31 | 46.42 |

Table 42: **Performance of Resa-STILL-4th-Layer** Each epoch contains 1448 Steps.

| CHECKPOINT STEPS | AIME24 | AIME25 | AMC23 | MATH500 | GPQA | MINERVA | AVG. |
|---|---|---|---|---|---|---|---|
| 1000 | 26.67 | 20.00 | 75.00 | 80.80 | 29.41 | 32.32 | 44.03 |
| 1500 | 30.00 | 16.67 | 72.50 | 82.80 | 30.51 | 30.81 | 43.88 |
| 2000 | 40.00 | 23.33 | 70.00 | 83.20 | 26.84 | 44.95 | 48.05 |
| 2500 | 33.33 | 26.67 | 70.00 | 83.60 | 29.41 | 31.31 | 45.72 |

Table 43: **Performance of Resa-STILL-5th-Layer** Each epoch contains 1448 Steps.

| CHECKPOINT STEPS | AIME24 | AIME25 | AMC23 | MATH500 | GPQA | MINERVA | AVG. |
|---|---|---|---|---|---|---|---|
| 1000 | 20.00 | 20.00 | 75.00 | 85.60 | 27.57 | 39.90 | 44.68 |
| 1500 | 30.00 | 20.00 | 80.00 | 83.20 | 30.88 | 31.31 | 45.90 |
| 2000 | 33.33 | 23.33 | 72.50 | 83.40 | 31.62 | 35.35 | 46.59 |
| 2500 | 30.00 | 16.67 | 72.50 | 82.60 | 27.94 | 38.38 | 44.68 |

Table 44: **Performance of Resa-STILL-6th-Layer** Each epoch contains 1448 Steps.

| CHECKPOINT STEPS | AIME24 | AIME25 | AMC23 | MATH500 | GPQA | MINERVA | AVG. |
|---|---|---|---|---|---|---|---|
| 1000 | 33.33 | 20.00 | 72.50 | 83.40 | 31.62 | 30.30 | 45.19 |
| 1500 | 26.67 | 26.67 | 77.50 | 81.60 | 27.94 | 35.86 | 46.04 |
| 2000 | 23.33 | 20.00 | 70.00 | 82.60 | 29.78 | 40.91 | 44.44 |
| 2500 | 30.00 | 33.33 | 67.50 | 83.20 | 24.63 | 36.36 | 45.84 |

Table 45: **Performance of Resa-STILL-7th-Layer** Each epoch contains 1448 Steps.

| CHECKPOINT STEPS | AIME24 | AIME25 | AMC23 | MATH500 | GPQA | MINERVA | AVG. |
|---|---|---|---|---|---|---|---|
| 1000 | 20.00 | 23.33 | 82.50 | 85.20 | 29.78 | 33.33 | 45.69 |
| 1500 | 16.67 | 23.33 | 70.00 | 82.60 | 27.21 | 34.34 | 42.36 |
| 2000 | 30.00 | 20.00 | 70.00 | 83.20 | 28.31 | 37.88 | 44.90 |
| 2500 | 16.67 | 20.00 | 72.50 | 81.00 | 29.04 | 28.79 | 41.33 |

Table 46: **Performance of Resa-STILL-8th-Layer** Each epoch contains 1448 Steps.

| CHECKPOINT STEPS | AIME24 | AIME25 | AMC23 | MATH500 | GPQA | MINERVA | AVG. |
|---|---|---|---|---|---|---|---|
| 1000 | 20.00 | 36.67 | 77.50 | 84.20 | 25.37 | 37.88 | 46.94 |
| 1500 | 36.67 | 20.00 | 75.00 | 83.80 | 28.31 | 34.34 | 46.35 |
| 2000 | 36.67 | 20.00 | 80.00 | 84.20 | 27.57 | 34.34 | 47.13 |
| 2500 | 26.67 | 26.67 | 67.50 | 83.80 | 27.57 | 38.89 | 45.18 |

Table 47: **Performance of Resa-STILL-9th-Layer** Each epoch contains 1448 Steps.

| CHECKPOINT STEPS | AIME24 | AIME25 | AMC23 | MATH500 | GPQA | MINERVA | AVG. |
|---|---|---|---|---|---|---|---|
| 1000 | 26.67 | 23.33 | 75.00 | 84.60 | 27.21 | 33.84 | 45.11 |
| 1500 | 36.67 | 23.33 | 67.50 | 84.80 | 29.78 | 37.37 | 46.58 |
| 2000 | 23.33 | 23.33 | 75.00 | 84.20 | 29.41 | 25.25 | 43.42 |
| 2500 | 36.67 | 23.33 | 70.00 | 83.20 | 29.41 | 36.36 | 46.50 |

Table 48: **Performance of Resa-STILL-10th-Layer** Each epoch contains 1448 Steps.

| CHECKPOINT STEPS | AIME24 | AIME25 | AMC23 | MATH500 | GPQA | MINERVA | AVG. |
|---|---|---|---|---|---|---|---|
| 1000 | 26.67 | 36.67 | 77.50 | 83.80 | 31.25 | 32.83 | 48.12 |
| 1500 | 26.67 | 33.33 | 75.00 | 84.60 | 30.51 | 34.34 | 47.41 |
| 2000 | 30.00 | 23.33 | 67.50 | 84.60 | 31.99 | 45.45 | 47.15 |
| 2500 | 20.00 | 26.67 | 72.50 | 83.60 | 33.09 | 33.84 | 44.95 |

Table 49: **Performance of Resa-STILL-11th-Layer** Each epoch contains 1448 Steps.

| CHECKPOINT STEPS | AIME24 | AIME25 | AMC23 | MATH500 | GPQA | MINERVA | AVG. |
|---|---|---|---|---|---|---|---|
| 1000 | 36.67 | 23.33 | 70.00 | 82.20 | 29.04 | 34.34 | 45.93 |
| 1500 | 20.00 | 23.33 | 72.50 | 84.60 | 26.84 | 40.40 | 44.61 |
| 2000 | 30.00 | 30.00 | 72.50 | 82.60 | 30.51 | 32.83 | 46.41 |
| 2500 | 30.00 | 26.67 | 77.50 | 84.60 | 31.99 | 34.85 | 47.60 |

Table 50: **Performance of Resa-STILL-12th-Layer** Each epoch contains 1448 Steps.

| CHECKPOINT STEPS | AIME24 | AIME25 | AMC23 | MATH500 | GPQA | MINERVA | AVG. |
|---|---|---|---|---|---|---|---|
| 1000 | 20.00 | 33.33 | 75.00 | 82.60 | 29.78 | 33.33 | 45.67 |
| 1500 | 33.33 | 23.33 | 75.00 | 82.80 | 30.88 | 30.81 | 46.03 |
| 2000 | 33.33 | 33.33 | 75.00 | 83.80 | 29.41 | 28.79 | 47.28 |
| 2500 | 30.00 | 23.33 | 77.50 | 84.20 | 26.47 | 33.33 | 45.81 |

Table 51: **Performance of Resa-STILL-13th-Layer** Each epoch contains 1448 Steps.

| CHECKPOINT STEPS | AIME24 | AIME25 | AMC23 | MATH500 | GPQA | MINERVA | AVG. |
|---|---|---|---|---|---|---|---|
| 1000 | 33.33 | 20.00 | 77.50 | 83.60 | 28.68 | 29.80 | 45.48 |
| 1500 | 26.67 | 26.67 | 70.00 | 84.80 | 26.84 | 30.81 | 44.30 |
| 2000 | 23.33 | 16.67 | 72.50 | 83.80 | 28.31 | 34.34 | 43.16 |
| 2500 | 33.33 | 23.33 | 65.00 | 82.80 | 30.88 | 32.32 | 44.61 |

Table 52: **Performance of Resa-STILL-14th-Layer** Each epoch contains 1448 Steps.

| CHECKPOINT STEPS | AIME24 | AIME25 | AMC23 | MATH500 | GPQA | MINERVA | AVG. |
|---|---|---|---|---|---|---|---|
| 1000 | 36.67 | 23.33 | 80.00 | 84.80 | 30.15 | 38.89 | 48.97 |
| 1500 | 26.67 | 26.67 | 72.50 | 84.40 | 30.51 | 31.82 | 45.43 |
| 2000 | 36.67 | 20.00 | 72.50 | 82.00 | 29.04 | 36.36 | 46.10 |
| 2500 | 30.00 | 16.67 | 67.50 | 83.80 | 32.72 | 36.87 | 44.59 |

Table 53: **Performance of Resa-STILL-15th-Layer** Each epoch contains 1448 Steps.

| CHECKPOINT STEPS | AIME24 | AIME25 | AMC23 | MATH500 | GPQA | MINERVA | AVG. |
|---|---|---|---|---|---|---|---|
| 1000 | 23.33 | 16.67 | 72.50 | 85.40 | 30.15 | 38.89 | 44.49 |
| 1500 | 30.00 | 20.00 | 65.00 | 83.40 | 29.78 | 32.83 | 43.50 |
| 2000 | 30.00 | 20.00 | 80.00 | 85.00 | 30.15 | 34.85 | 46.67 |
| 2500 | 30.00 | 20.00 | 70.00 | 82.80 | 30.15 | 38.38 | 45.22 |

Table 54: **Performance of Resa-STILL-16th-Layer** Each epoch contains 1448 Steps.

| CHECKPOINT STEPS | AIME24 | AIME25 | AMC23 | MATH500 | GPQA | MINERVA | AVG. |
|---|---|---|---|---|---|---|---|
| 1000 | 33.33 | 30.00 | 67.50 | 85.40 | 30.88 | 36.36 | 47.25 |
| 1500 | 33.33 | 13.33 | 67.50 | 83.00 | 30.88 | 35.86 | 43.98 |
| 2000 | 36.67 | 23.33 | 65.00 | 83.20 | 31.25 | 34.85 | 45.72 |
| 2500 | 43.33 | 16.67 | 77.50 | 83.80 | 30.88 | 36.36 | 48.09 |

Table 55: **Performance of Resa-STILL-17th-Layer** Each epoch contains 1448 Steps.

| CHECKPOINT STEPS | AIME24 | AIME25 | AMC23 | MATH500 | GPQA | MINERVA | AVG. |
|---|---|---|---|---|---|---|---|
| 1000 | 43.33 | 20.00 | 77.50 | 84.00 | 33.82 | 37.88 | 49.42 |
| 1500 | 26.67 | 20.00 | 75.00 | 83.00 | 29.41 | 35.86 | 44.99 |
| 2000 | 23.33 | 23.33 | 75.00 | 83.40 | 33.82 | 38.38 | 46.21 |
| 2500 | 43.33 | 23.33 | 72.50 | 84.40 | 29.04 | 31.82 | 47.40 |

Table 56: **Performance of Resa-STILL-18th-Layer** Each epoch contains 1448 Steps.

| CHECKPOINT STEPS | AIME24 | AIME25 | AMC23 | MATH500 | GPQA | MINERVA | AVG. |
|---|---|---|---|---|---|---|---|
| 1000 | 40.00 | 13.33 | 75.00 | 84.20 | 27.94 | 38.38 | 46.48 |
| 1500 | 30.00 | 23.33 | 72.50 | 83.20 | 30.15 | 35.35 | 45.76 |
| 2000 | 33.33 | 30.00 | 72.50 | 84.20 | 29.04 | 36.87 | 47.66 |
| 2500 | 26.67 | 23.33 | 75.00 | 84.80 | 28.68 | 35.35 | 45.64 |

Table 57: **Performance of Resa-STILL-19th-Layer** Each epoch contains 1448 Steps.

| CHECKPOINT STEPS | AIME24 | AIME25 | AMC23 | MATH500 | GPQA | MINERVA | AVG. |
|---|---|---|---|---|---|---|---|
| 1000 | 26.67 | 33.33 | 70.00 | 83.60 | 27.21 | 31.82 | 45.44 |
| 1500 | 33.33 | 23.33 | 75.00 | 85.00 | 29.04 | 29.29 | 45.83 |
| 2000 | 20.00 | 30.00 | 65.00 | 83.60 | 28.31 | 27.27 | 42.36 |
| 2500 | 40.00 | 20.00 | 67.50 | 82.80 | 27.94 | 33.84 | 45.35 |

Table 58: **Performance of Resa-STILL-20th-Layer** Each epoch contains 1448 Steps.

| CHECKPOINT STEPS | AIME24 | AIME25 | AMC23 | MATH500 | GPQA | MINERVA | AVG. |
|---|---|---|---|---|---|---|---|
| 1000 | 30.00 | 33.33 | 75.00 | 84.60 | 27.57 | 36.87 | 47.90 |
| 1500 | 33.33 | 16.67 | 72.50 | 84.20 | 30.51 | 30.81 | 44.67 |
| 2000 | 40.00 | 20.00 | 62.50 | 81.80 | 29.41 | 34.85 | 44.76 |
| 2500 | 23.33 | 30.00 | 70.00 | 84.00 | 30.88 | 33.84 | 45.34 |

Table 59: **Performance of Resa-STILL-21st-Layer** Each epoch contains 1448 Steps.

| CHECKPOINT STEPS | AIME24 | AIME25 | AMC23 | MATH500 | GPQA | MINERVA | AVG. |
|---|---|---|---|---|---|---|---|
| 1000 | 33.33 | 23.33 | 75.00 | 82.20 | 31.99 | 34.34 | 46.70 |
| 1500 | 30.00 | 26.67 | 70.00 | 83.20 | 31.62 | 35.86 | 46.23 |
| 2000 | 23.33 | 16.67 | 70.00 | 83.80 | 31.25 | 34.85 | 43.32 |
| 2500 | 23.33 | 20.00 | 72.50 | 85.80 | 29.04 | 39.39 | 45.01 |

Table 60: **Performance of Resa-STILL-22nd-Layer** Each epoch contains 1448 Steps.

| CHECKPOINT STEPS | AIME24 | AIME25 | AMC23 | MATH500 | GPQA | MINERVA | AVG. |
|---|---|---|---|---|---|---|---|
| 1000 | 30.00 | 20.00 | 72.50 | 83.80 | 28.31 | 32.32 | 44.49 |
| 1500 | 30.00 | 20.00 | 80.00 | 82.00 | 29.04 | 35.35 | 46.07 |
| 2000 | 23.33 | 23.33 | 60.00 | 84.80 | 27.57 | 37.37 | 42.73 |
| 2500 | 23.33 | 23.33 | 67.50 | 85.00 | 29.78 | 43.43 | 45.40 |

Table 61: **Performance of Resa-STILL-23rd-Layer** Each epoch contains 1448 Steps.

| CHECKPOINT STEPS | AIME24 | AIME25 | AMC23 | MATH500 | GPQA | MINERVA | AVG. |
|---|---|---|---|---|---|---|---|
| 1000 | 36.67 | 20.00 | 77.50 | 83.80 | 30.51 | 29.80 | 46.38 |
| 1500 | 20.00 | 16.67 | 65.00 | 85.20 | 29.41 | 34.85 | 41.86 |
| 2000 | 26.67 | 33.33 | 67.50 | 85.40 | 30.15 | 33.33 | 46.06 |
| 2500 | 33.33 | 16.67 | 75.00 | 82.80 | 29.78 | 33.33 | 45.15 |

Table 62: **Performance of Resa-STILL-24th-Layer** Each epoch contains 1448 Steps.

| CHECKPOINT STEPS | AIME24 | AIME25 | AMC23 | MATH500 | GPQA | MINERVA | AVG. |
|---|---|---|---|---|---|---|---|
| 1000 | 40.00 | 30.00 | 70.00 | 84.20 | 28.68 | 37.88 | 48.46 |
| 1500 | 30.00 | 26.67 | 67.50 | 83.20 | 29.41 | 35.86 | 45.44 |
| 2000 | 23.33 | 16.67 | 75.00 | 82.80 | 30.88 | 36.36 | 44.17 |
| 2500 | 30.00 | 26.67 | 67.50 | 84.60 | 27.57 | 34.34 | 45.11 |

Table 63: **Performance of Resa-STILL-25th-Layer** Each epoch contains 1448 Steps.

| CHECKPOINT STEPS | AIME24 | AIME25 | AMC23 | MATH500 | GPQA | MINERVA | AVG. |
|---|---|---|---|---|---|---|---|
| 1000 | 36.67 | 20.00 | 65.00 | 85.60 | 30.88 | 36.87 | 45.84 |
| 1500 | 26.67 | 16.67 | 75.00 | 83.60 | 30.15 | 33.33 | 44.24 |
| 2000 | 26.67 | 23.33 | 67.50 | 84.40 | 27.21 | 35.86 | 44.16 |
| 2500 | 30.00 | 23.33 | 70.00 | 83.00 | 31.62 | 36.87 | 45.80 |

Table 64: **Performance of Resa-STILL-26th-Layer** Each epoch contains 1448 Steps.

| CHECKPOINT STEPS | AIME24 | AIME25 | AMC23 | MATH500 | GPQA | MINERVA | AVG. |
|---|---|---|---|---|---|---|---|
| 1000 | 16.67 | 26.67 | 65.00 | 82.80 | 30.51 | 34.85 | 42.75 |
| 1500 | 33.33 | 23.33 | 72.50 | 82.80 | 31.62 | 36.87 | 46.74 |
| 2000 | 36.67 | 30.00 | 62.50 | 83.00 | 27.94 | 33.84 | 45.66 |
| 2500 | 30.00 | 36.67 | 67.50 | 83.00 | 27.94 | 40.40 | 47.59 |

Table 65: **Performance of Resa-STILL-27th-Layer** Each epoch contains 1448 Steps.

