# OpenReview forum: "Resa: Efficient Reasoning Models via SAEs"
_ICLR.cc/2026/Conference — Submitted to ICLR 2026_

### Official Review · Reviewer_CEct · 2025-10-26

**Soundness:** 3
**Presentation:** 3
**Contribution:** 3
**Rating:** 6
**Confidence:** 3

**Summary:**

The paper presents SAE-Tuning, which first trains an SAE to capture reasoning abilities from a source model and then uses the trained SAE to guide a supervised finetuning process to elicit these abilities in a target model. The authors verify its effectiveness on two models and further show that the training can be simplified from both the source and SAE aspects. They also demonstrate that the reasoning abilities extracted via SAEs can be transferred across both data distributions and models.

**Strengths:**

1. The paper is well-written and easy to follow.
2. The SAE-Tuning method is novel and cost-effective compared to the RL method.

**Weaknesses:**

1. In the experiment, the target model is always an R1-distilled method. But from a feature extraction perspective, features should first be extracted from an R1-distilled model and further guide the tuning of a non-R1-like model. Therefore, the application of the method remains limited.
2. The experiments are limited to small models and math tasks. A larger scale and more tasks are required to verify its effectiveness.

**Questions:**

- How does the performance compare to the RL method when the target model is not R1-distilled?
- Have the authors tested on other tasks and model scales?
- How do the model's reasoning behaviours, such as response length, change when applying the SAE-Tuning?

---

> ### Author Response · Authors · 2025-11-21
>
> We thank the reviewer for the positive feedback on the clarity, novelty, and cost-effectiveness of our method. The reviewer raises thoughtful questions regarding architectural generalization, model/task scale, and behavioral effects. **We have updated our paper to remove confusion, improve content ordering and add new results to address them thoroughly (all updates in our revised paper are marked with blue text).**
>
> ### **Application to cross-architecture models**
>
> We agree that, in its current form, SAE-Tuning requires the source and target models to share the same hidden dimensions and transformer structure. This constraint arises because the frozen SAE must be inserted directly into the target model’s computation graph during SAE-guided SFT. As a result, in the current work, we did not evaluate transfers where the source is R1-based and the target is architecturally unrelated.
>
> At the same time, we would like to clarify the forms of generalization that are already present:
>
> - **Architecture-agnostic mechanism.** We show that SAE-Tuning itself is not tied to a single model family. The full pipeline works cleanly on Qwen-style models (Section 4.1) and on Llama-style models (Section 4.2), where the target becomes OctoThinker-3B. This demonstrates that SAE-Tuning is operational across different architectural families when trained natively within each family.
> - **Flexible feature extraction within a family.** In Section 5, we extract reasoning abilities from Qwen, Qwen-Math, and R1-Distill models and successfully apply these as modular adapters to R1-Distill. Within-family extraction and transfer do not require that the source be R1-Distilled specifically.
>
> We agree that true cross-architecture transfer (e.g., injecting R1-derived reasoning into a non-R1-like model) is an important direction for future work. Supporting this will likely require a learned mapping between hidden spaces (e.g., a transcoder), and we now state this limitation explicitly.
>
> ### **Extensions of Resa to additional models and tasks**
>
> We agree with the reviewer that the original submission focused on smaller models. In the revised paper, we now include new experiments on 8B-parameter Llama-style models (OctoThinker-8B; Table 7), which show that SAE-Tuning continues to yield consistent gains at a larger scale. These results complement our earlier experiments on 1.5B and 3B Qwen-style models and demonstrate that the method applies across both model families and model sizes.
> Regarding task diversity, our evaluation primarily targets math and scientific reasoning benchmarks (AIME, AMC, MATH500, GPQA, Minerva), where correctness is objective and widely used in prior RL-based reasoning work. We agree that extending SAE-Tuning to additional domains—such as coding, general logical reasoning, or open-ended QA—is an important next step for establishing broader generality. We now highlight this as a direction for future work.
>
> ### **Changes in reasoning behavior**
>
> Although our SAE-Tuning procedure is CoT-free (meaning it does not use step-by-step reasoning traces), we noticed that the model's reasoning behavior, when prompted to generate a response, evolves in a way that is qualitatively similar to standard RL post-training. Specifically, we observed that the average response length tends to increase during the initial phase of SAE-Tuning and then converges as training progresses.
>
> We thank the reviewer again for the constructive and encouraging assessment. Their comments have helped us clarify the scope and limitations of the current work and refine the presentation of our contributions.

---

> > ### Comment · Reviewer_CEct · 2025-11-23
> > **Official Comment by Reviewer CEct**
> >
> > Thank you for your response. Regarding W1, I was not referring to architectural differences between the source and target models. Instead, I meant that the source model possesses Long-CoT reasoning capabilities, while the target model does not. The question is: can feature extraction help transfer Long-CoT reasoning from the source to the target? Architectural differences are not required for this process.

---

> > > ### Author Response · Authors · 2025-11-26
> > >
> > > ### **Long-CoT Transfer**
> > >
> > > We thank the reviewer for the clarification. To directly address the question—whether SAE-Tuning can transfer Long-CoT reasoning from a stronger (e.g., R1-distilled) source model into a target model that does not have Long-CoT capabilities—we conducted an additional experiment using R1-Distill as the source and Qwen2.5-Math-1.5B as the target.
> > >
> > > | | AIME24 | AIME25 | AMC23 | MATH500 | GPQA | Minerva |
> > > | :--- | :--- | :--- | :--- | :--- | :--- | :--- |
> > > | Qwen2.5-Math-1.5B | 3.33 | 3.33 | 35 | 42.6 | 22.72 | 13.23 |
> > > | R1-Distill-SAE-Tune-Qwen2.5-Math-1.5B | 6.67 | 3.33 | 40 | 42.4 | 30.8 | 12.5 |
> > >
> > > These results show that SAE-Tuning successfully transfers format reasoning behavior to a non-R1-Distilled target, improving AMC23, GPQA, and AIME24. The gains are naturally smaller than when the target already has strong reasoning scaffolding (e.g., R1-Distill), as Qwen-Math lacks some of the underlying knowledge. Nevertheless, the experiment confirms the reviewer’s intuition: SAE-Tuning can transfer Long-CoT reasoning features even when the target model does not initially possess them.

---

> > > > ### Comment · Reviewer_CEct · 2025-11-27
> > > > **Official Comment by Reviewer CEct**
> > > >
> > > > Thanks for your response! I have raised the score.

---

### Official Review · Reviewer_CHYk · 2025-10-30

**Soundness:** 2
**Presentation:** 2
**Contribution:** 3
**Rating:** 4
**Confidence:** 5

**Summary:**

This paper introduces **Resa**, a family of reasoning models trained via **SAE-Tuning** (Sparse Autoencoder Tuning), a novel two-stage method that elicits reasoning abilities in language models without requiring reinforcement learning or chain-of-thought traces. The method first trains a sparse autoencoder (SAE) to extract latent reasoning features from a source model's activations, then inserts the frozen SAE into a target model to guide supervised fine-tuning with LoRA adapters using only verified question-answer pairs. The key contributions include: (1) achieving comparable performance to RL-trained models while reducing training costs by 2000× (to $1) and time by 450× (20 minutes), (2) demonstrating that reasoning abilities extracted via SAEs are generalizable across datasets and modular across models within the same family, working as portable "adapters" at inference time, and (3) showing effectiveness across model architectures (Qwen and Llama), with 1.5B models achieving 43.33% Pass@1 on AIME24 and 90% on AMC23, suggesting that reasoning abilities exist latently in base models and can be efficiently extracted and transferred.

**Strengths:**

### Originality
The paper presents a novel application of sparse autoencoders to reasoning ability transfer, which is creative and distinct from prior work. While SAEs have been used for interpretability and steering, using them as a bridge to extract and transfer reasoning capabilities between models is innovative. The concept of treating reasoning as a "modular adapter" that can be plugged into models at test time without retraining is particularly original and opens new research directions.

### Quality
The experimental work is comprehensive, with extensive ablations across multiple dimensions (SAE training modes, source models, layer selection, datasets). The paper evaluates on six diverse reasoning benchmarks and demonstrates consistency across different model families (Qwen and Llama). The inclusion of detailed cost and time analysis (Table 6) strengthens the practical claims. The layerwise analysis (Section C.1) with GMM fitting provides interesting insights into where reasoning features reside in the model.

**Weaknesses:**

## 1. Insufficient Evidence for Core Theoretical Claim

**Lines 46-48:** The paper claims "within this dictionary, certain features must correspond to the fundamental building blocks of reasoning" as a key insight. However, no empirical evidence or theoretical justification is provided to support this claim. This is a foundational assumption for the entire SAE-Tuning methodology, yet it remains unsubstantiated. The authors should provide either:
- Quantitative analysis showing which SAE features activate during reasoning tasks
- Ablation studies demonstrating that specific features are necessary for reasoning performance
- Theoretical arguments grounded in interpretability literature

## 2. Unfair Experimental Comparison in Ablation Study

**Lines 315-318, Table 1:** The comparison between Resa-STILL-v1 and STILL-CoT-free-SFT is fundamentally flawed:
- **Resa-STILL-v1** is trained using SAE-Tuning with Tina-STILL as the source model, where Tina-STILL has already acquired long CoT capabilities through RL training
- **STILL-CoT-free-SFT** appears to be trained directly from STILL-3-1.5B-preview (44.86% avg), which lacks long CoT capabilities
- This creates an unfair comparison: Resa inherits reasoning abilities from an RL-trained model, while the baseline starts from a weaker foundation
- More puzzlingly, STILL-CoT-free-SFT (39.00%) performs *worse* than its presumed base model STILL-3-1.5B-preview (44.86%), raising questions about training stability or base model identity

**Required clarification:**
- What is the exact base model for STILL-CoT-free-SFT?
- Why does training decrease performance?
- A fair comparison would use the same base model (STILL-3-1.5B-preview)

## 3. Conceptual Inconsistency in SAE Training Modes

**Lines 192-196:** The "Pre-trained" SAE mode directly contradicts the paper's core methodology:
- The paper emphasizes that Stage I is essential for SAEs to learn reasoning features from the source model on the trigger dataset
- However, the "Pre-trained" mode uses a generic SAE trained on R1-Distill and "bypasses Stage I entirely"
- This raises fundamental questions: How can a pre-trained SAE that has never seen the source model (e.g., Tina-STILL) or the trigger dataset capture source-specific reasoning features?
- Table 2 shows Pre-trained SAE performs worse (44.99%) than Fine-tuned (47.28%) or Trained-from-Scratch (47.36%), but the fact that it works at all contradicts the claimed necessity of Stage I

**This inconsistency undermines the paper's theoretical framework and requires clarification.**

## 4. Minor Typographical Error

**Line 216 (Equation 3):** "where Top-k means that we only the top k features in the vector" is missing a verb. Should read "we only **keep** the top k features" or similar.

## 5. Incomplete Evaluation Protocol Documentation

**Throughout the paper:** Critical evaluation details are missing or buried in the appendix:
- The evaluation metric (Pass@1) is only mentioned on line 755 in the appendix, not in the main text or table captions
- **Sampling details are completely absent:** How many times is each problem sampled? What are the sampling parameters (temperature, top-p)?
- For reasoning benchmarks, these details significantly impact reproducibility and fair comparison
- This information should be prominently displayed in Section 4 or in table captions

Although I am not an expert in this domain, I think the writing could be clearer to help readers unfamiliar with the topic better understand your paper. If your response is reasonable, I am happy to raise my score.


## 6. Misleading Cost Claims Due to Upstream RL Dependency

**Core issue:** The paper's central claim of 2000× cost reduction ($2268 → $1) is misleading because it **does not account for the upstream RL training costs** required to create the source models.

**The dependency chain:**
- SAE-Tuning extracts reasoning abilities from source models like Tina-STILL or R1-Distill
- **Tina-STILL** was trained via expensive RL from R1-Distill
- **R1-Distill** itself was created by distilling reasoning traces from DeepSeek-R1, which required expensive RL training
- Therefore, the complete cost is: **Upstream RL cost (thousands of dollars) + SAE-Tuning ($1)**

**Why this matters:**
- SAE-Tuning does **not create reasoning abilities from scratch**—it only transfers/copies existing abilities from RL-trained models
- Without access to an RL-trained source model with long CoT capabilities, the method cannot work
- The claimed "$1 reasoning model" is actually "$1 to copy an existing reasoning model that cost thousands to create"
- This is fundamentally a knowledge distillation/transfer method, not a standalone reasoning elicitation method

**Fair framing:** The paper should position SAE-Tuning as an efficient method for **transferring** reasoning abilities between models (assuming one RL-trained model exists), rather than claiming it eliminates the need for RL training entirely. The cost comparison should either include amortized upstream costs or be clearly framed as "marginal cost per additional model."

**Questions:**

## Q1: Evidence for Reasoning Feature Identification
Can you provide concrete evidence that SAE features correspond to reasoning building blocks?

## Q2: Clarification on Ablation Baseline
For the STILL-CoT-free-SFT baseline in Table 1:
- What is the exact base model used for this baseline?
- If it's STILL-3-1.5B-preview, why does performance decrease from 44.86% to 39.00% after training?
- Can you provide a fair comparison where both Resa and the baseline start from the same base model?
- What was the training setup (learning rate, epochs, etc.) that led to this performance degradation?

## Q3: Pre-trained SAE Mode Mechanism
Regarding the "Pre-trained" SAE mode:
- How can a pre-trained SAE that bypasses Stage I and never sees the source model or trigger dataset still extract reasoning abilities?

## Q4: Detailed Evaluation Protocol
Can you provide complete evaluation details:
- How many samples per problem for Pass@1? (Is it truly 1 sample per problem, or multiple samples with pass@1 criterion?)
- What are the exact sampling parameters (temperature, top-p, etc.)?
- Are these consistent across all baselines and your method?
- Why was this information relegated to the appendix rather than included in the main experimental section?

---

> ### Author Response · Authors · 2025-11-21
> **Response to CHYk - Part 1**
>
> We sincerely thank the reviewer for the detailed, thoughtful, and constructive comments. We are particularly grateful that you found the use of SAEs for reasoning transfer to be “creative,” “distinct,” and “innovative,” and that you regard the experiments and analysis as comprehensive and high quality.
>
> Your feedback helped us identify several key ambiguities in our initial framing. **We have updated our paper to remove confusion, improve content ordering and add new results to address them thoroughly (all updates in our revised paper are marked with blue text).** Particularly, in the revised version, we have clarified the core theoretical claim, baselines, cost framing, and evaluation setup.
>
> ### **Evidence for the “building blocks of reasoning” claim**
>
> We agree that our original wording (“must correspond to the fundamental building blocks of reasoning”) was too strong without sufficient justification. In the revised paper, we soften this statement and frame it as a hypothesis supported by preliminary evidence rather than as a proved fact. We now explicitly present two lines of evidence:
>
> **Ablation across SAE training modes (Table 2)**
>
> - A pre-trained SAE (trained on generic text) improves R1-Distill from 41.18% to 44.99%. A trained-from-scratch or fine-tuned SAE on the reasoning trigger dataset (STILL) yields substantially higher performance (47.36% to 48.06%).
> - Given that all variants use the same LoRA-based SFT and the same CoT-free QA data, the additional gain from task-specific SAE training strongly suggests that the SAE is learning features aligned with reasoning on the trigger distribution, not just acting as a generic bottleneck.
>
> **Layerwise “reasoning feature” analysis (Appendix C.1)**
>
> - We define a proxy for “reasoning features” as SAE features that activate exclusively and simultaneously at \<think\> and \</think\> tokens in a standard reasoning prompt, and not at other positions.
> - For each layer, we count such features in the base model and compare this distribution to the final performance of 26 Resa-STILL models where the SAE is hooked at different layers.
> - Fitting a 3-component Gaussian mixture model (GMM) to both distributions reveals closely aligned cluster means, weights, and entropy, indicating that layers with richer “thinking-token-only” features tend to yield higher downstream reasoning performance under SAE-Tuning.
>
> We present these as correlative rather than causal evidence and explicitly note that fully identifying “fundamental building blocks of reasoning” at the feature level is an open challenge for interpretability. Our contribution is to provide a concrete, empirically validated procedure that isolates a subset of SAE features strongly tied to reasoning performance and to propose these as a promising substrate for future mechanistic analysis.

---

> ### Author Response · Authors · 2025-11-21
> **Response to CHYk - Part 2**
>
> ### **Ablation baseline and fairness of comparison**
>
> We agree that our original exposition around Table 1 and the STILL-CoT-free-SFT baseline was confusing.
>
> **Base model clarification.** In the revised version, we clarify that:
>
> - The base model for the STILL-CoT-free-SFT baseline is R1-Distill, whose average performance is 41.18%.
> - STILL-3-1.5B-preview (44.86%) is a separate RL-trained model used as a reference, not as the base for the SFT baseline.
>
> Thus, the fair comparison that the reviewer requested (“same base model”) is already present:
>
> - Baseline (standard SFT from R1-Distill): R1-Distill → STILL-CoT-free-SFT: 41.18% → 39.00%
> - Our method (SAE-Tuning from R1-Distill): R1-Distill → Resa-STILL-v5: 41.18% → 48.06%
>
> We now make this alignment explicit in Section 4 and the Table 1 caption.
>
> **Why does STILL-CoT-free-SFT degrade performance?** This behavior deserves explanation. The CoT-free QA data are highly specialized and differ from the more diverse distributions on which R1-Distill was trained. Our interpretation is that naive SFT on these answers induces catastrophic forgetting of useful behaviors without successfully eliciting stronger reasoning. This baseline is intentionally included to illustrate that:
>
> - Simply fine-tuning on final answers is not enough to elicit reasoning, and
> - The SAE-guided structure is critical to making CoT-free training effective.
>
> In the modified paper, we highlight the failure of STILL-CoT-free-SFT in Appendix C.1 as an important negative result motivating SAE-Tuning.
>
> ### **Pre-trained SAE mode and the role of stage I**
>
> We acknowledg that the original description gave the impression that Stage I is absolutely required for SAE-Tuning to work, which seems at odds with the fact that the pre-trained SAE mode also shows some improvement. We have clarified the intended interpretation:
>
> - The pre-trained SAE is trained on R1-Distill activations from a large, generic text corpus (not on the STILL trigger dataset and not on Tina-STILL).
> - When inserted into the model and used as a frozen bottleneck, it acts as a generic sparse representation constraint: it reshapes activations into a structured feature space that is somewhat helpful even without being specialized to the trigger data. This explains why Resa-STILL-v2 (pre-trained SAE) still improves over R1-Distill (44.99% vs. 41.18%).
>
> However, the key empirical fact is that:
>
> - Pre-trained SAE: 44.99%
> - Fine-tuned / from-scratch SAE on STILL: 47.28% and 48.06%
>
> We now explicitly state that:
>
> - Stage I is not necessary for SAE-Tuning to have any effect, but
> - Stage I is essential to obtain optimal reasoning performance, precisely because it allows the SAE to adapt its dictionary to reasoning-specific activations from the trigger dataset.
>
> In other words, the pre-trained SAE ablation is not a contradiction; it is evidence that the curriculum-specific Stage I training is what converts a generic sparse dictionary into an effective “reasoning teacher.”
>
> ### **Other questions**
>
> **Typographical Error.** Thank you for catching this. We have corrected the sentence in Line 478 in our final draft to: "...we only keep the top k features in the vector."
>
> **Incomplete Evaluation Protocol.** This information is now in the main text. In the modified version, we have moved all evaluation details to Section 4, Evaluation Setup.
>
> - Metric: All scores are zero-shot Pass@1 Mean@10.
> - Sampling parameters: We use a temperature of 0.6 and top-p of 0.95.
> - Consistency: These parameters are applied via a standardized setup across all evaluated models for a fair comparison.
>
> We are grateful for the opportunity to improve our work based on this constructive feedback and would be happy to provide any further clarification.

---

### Official Review · Reviewer_2vob · 2025-10-31

**Soundness:** 2
**Presentation:** 3
**Contribution:** 3
**Rating:** 4
**Confidence:** 3

**Summary:**

This paper introduces RESA, a method for eliciting reasoning ability in language models using Sparse Autoencoder (SAE) tuning.
The key idea is to extract latent reasoning features from a teacher model (often an RL-trained reasoning model like Tina) via an SAE, and then inject those features into a target model using a lightweight SFT procedure.
The authors claim that this achieves near-RL-level reasoning performance at a fraction of the cost and without reinforcement learning or Chain-of-Thought (CoT) data.

**Strengths:**

1.Timely and interesting topic.

The paper targets an important direction: efficient reasoning ability transfer.
It goes beyond standard output-level distillation by exploring structural feature transfer through SAEs — a refreshing and potentially impactful angle.
Indeed, model capability acquisition is not limited to simple knowledge distillation, and studying low-cost, structural-level distillation is both novel and valuable.

2.Good integration of interpretability and model training.
Using SAEs to capture internal reasoning representations bridges interpretability and capability transfer.
This is a strong conceptual contribution that could inspire future work at this intersection.

3.Clear writing and presentation.
The paper is well organized and generally easy to follow.
Figures and tables clearly convey the main experimental findings.

**Weaknesses:**

1. Questionable claim of “efficiency.”

While the paper repeatedly claims RESA is efficient, the evidence is not fully convincing.
  - Section 4.1 only shows capability replication, not true efficiency.  RESA is demonstrated to replicate reasoning performance of an RL-trained model (Tina) using much cheaper fine-tuning,  but this assumes the existence of that RL teacher. Training Tina itself is extremely costly. Therefore, if we count the full pipeline cost (RL teacher + RESA transfer), RESA is not cheaper overall — it is conditionally efficient only when a high-quality teacher already exists. In contrast, Section 4.2 includes a self-distillation experiment, which also achieves reasonably strong performance, suggesting that RESA may not necessarily rely on an expensive RL teacher to obtain comparable results？
  - The “self-teacher” experiment (§4.2) partially addresses this but with misaligned settings. The Tina teacher and the R1-Distill teacher are trained on different data and objectives. Specifically, in the Tina pipeline, the teacher used for distillation is the same model trained on the corresponding dataset (e.g., Tina-STILL’s teacher is the model produce the STILL data). In contrast, RESA uses R1-Distill-Qwen-1.5B as its teacher across all experiments, which was not trained on the same dataset or objective as Tina’s teacher. Thus, their comparison does not isolate the gain from SAE-Tuning alone. A fairer efficiency claim should compare RESA vs. standard distillation.For example: given the same teacher, same base model, and same QA data, one could compare (a) traditional distillation (prompt–response SFT) vs. (b) RESA’s SAE-guided tuning. This would show whether RESA is not only cheaper than RL, but also more data-efficient or optimization-efficient than conventional post-training.
- Moreover, in practice, if a strong checkpoint already exists, directly using that model remains the most cost-effective option, making the claimed “efficiency” of RESA less meaningful in realistic scenarios.

2. Lack of discussion or evidence on scalability.
  - No evidence of efficiency in scaling direction. “Efficiency” would be more convincing if the method could transfer reasoning from a large teacher (e.g., 7B) to a smaller model (e.g., 1.5B) with reduced cost. Such large-to-small transfer is the true motivation behind efficiency, yet no such experiment is included. Overall, Section 4.1 validates feasibility (“it works”) but not efficiency (“it works better”).
  -  Although RESA’s optimization cost is small (only LoRA adapters are trained), the SAE module itself is extremely large: For a 1.5B base model (hidden size 4096), the SAE has roughly 2.1B parameters. Since SAE scales as O(d^2) with hidden size, applying it to 7B–70B models would require billions to tens of billions of parameters just for the SAE. This raises major scalability questions:
    - Can such an SAE be trained or even stored for larger models?
    - Does the efficiency still hold when the feature dimension grows?
    - Is there a way to share or compress SAEs across layers or models?
Currently, all experiments are limited to 1.5B models, and scalability is only discussed conceptually.
Hence, the method’s practicality for modern large-scale models remains unclear.

**Questions:**

1. On the Efficiency Claim

- How should the claimed “efficiency” be interpreted when training the RL teacher (Tina) is itself very costly?
- The self-distillation experiment (§4.2) performs well without an RL teacher. Does this imply that RESA’s efficiency mainly comes from self-distillation rather than teacher-based distillation?
- Have you compared RESA with standard distillation or SFT under identical settings (same teacher, same base model, same QA data)? Such a comparison would help validate whether RESA is truly more efficient or simply an alternative distillation form.

2. Scalability of the SAE Module

- How does the parameter count of the SAE grow with model size? For example, with a hidden size of 4096 (1.5B model), the SAE already contains ~2.1B parameters. Would applying RESA to 7B or larger models require prohibitively large SAEs?

---

> ### Author Response · Authors · 2025-11-21
>
> We thank the reviewer for the constructive feedback and for highlighting the novelty of using SAEs to bridge interpretability and capability transfer. We appreciate the reviewer’s positive assessment of the conceptual contribution, presentation clarity, and the motivation for studying low-cost structural distillation. The reviewer raises two important questions regarding our (1) interpretation of efficiency and (2) scalability of the SAE module. We fully agree that these are central issues **and we have updated our paper to remove confusion, improve content ordering and add new results to address them thoroughly (all updates in our revised paper are marked with blue text).**
>
> ### **The "efficiency" claim**
>
> **Clarifying what "efficiency" refers to.** We agree that efficiency should not rely on the existence of an expensive RL teacher. Our primary claim is not that RESA replicates a costly RL model cheaply, but that the entire RL post-training step can be replaced with our lightweight ($1, ~20 min) SAE-Tuning procedure applied directly to the base model.
>
> This is demonstrated by Resa-STILL-v5 (Table 2), which uses:
>
> - Source model: R1-Distill (base model)
> - Target model: R1-Distill
> - No RL teacher
> - Trained-from-scratch SAE
>
> Resa-STILL-v5 reaches 48.06% average Pass@1—matching the RL-trained Tina-STILL model (48.16%) while avoiding the RL stage entirely. We have emphasized this result more clearly in the revised paper. This directly addresses Q1: the efficiency claim concerns removing the expensive RL post-training pipeline, not copying its output.
>
> **Does efficiency come from “self-distillation”?** The reviewer correctly observes that Stage II can be seen as a form of self-distillation. The key question (Q2) is whether SAE-Tuning is genuinely more effective than standard SFT under identical conditions. We conducted exactly this comparison:
>
> - Standard SFT baseline (same teacher, same data, same base model): STILL-CoT-free-SFT → 39.00%
> - Our method (same teacher, same data, same base model): Resa-STILL-v5 → 48.06%
>
> Thus, the improvement cannot be attributed merely to distillation—it comes from the SAE-guided mechanism, which successfully elicits latent reasoning abilities where standard SFT does not. We highlight this contrast more explicitly in the revision.
>
> Regarding the reviewer’s concern about teacher heterogeneity: we agree that the original draft over-emphasized capability replication (using Tina), which could obscure the core contribution. The revised version now clearly centers the base-model elicitation result.
>
> **On Practicality When a Strong Teacher Exists.** We agree with the reviewer that, in practice, if you already have a strong existing RL model, then no additional tuning/training is needed. However, our contribution is to provide a new training method when such a teacher is unavailable or costly to obtain. In addition, Section 5 shows that SAE-Tuning produces modular reasoning adapters that can be attached to other models, offering flexibility beyond a monolithic RL checkpoint.
>
> ### **Scalability of the SAE module**
>
> We appreciate the reviewer raising this point. It is an important open question.
>
> **On the Parameter Count Calculation.** The reviewer’s parameter estimate assumed a hidden size of 4096. The Qwen-1.5B/R1-Distill model uses 1536 hidden units.
>
> - The Qwen-1.5B model (R1-Distill) has a hidden size ($d$) of 1536.
> - Our SAE (Table 6) uses 65,536 features ($m$).
> - The SAE consists of an encoder $W_{enc} \in \mathbb{R}^{m \times d}$ and a decoder $W_{dec} \in \mathbb{R}^{d \times m}$.
> - The total parameter count is $2 \times d \times m = 2 \times 1536 \times 65536 \approx$ 201 Million parameters.
>
> This is significant, but it is ~13% of the 1.5B base model's size, not the 2.1B parameters (which would be >140% of the model) estimated by the reviewer. This size is well within feasible training and storage limits.
>
> **On the General Scalability Challenge.** The reviewer's underlying point is valid. Scaling this to a 70B model with the same expansion factor would be a challenge. We agree this is a key limitation and a crucial area for future work. As the reviewer suggests, exploring SAE parameter sharing across layers, SAE compression, or optimizing expansion factors are all vital next steps to apply this method to larger models.
>
> **On Large-to-Small Transfer.** We agree that large-to-small transfer is an important test of efficiency. Our initial scope was to establish that SAE-Tuning itself works reliably and performs latent elicitation on base models. Supporting heterogeneous hidden sizes (e.g., 7B–>1.5B) requires an intermediate mapping mechanism (e.g., a transcoder). We have now emphasized this in the future directions section.
>
> We hope these clarifications address the reviewer's concerns and properly frame our contribution as an efficient, RL-free method for eliciting latent reasoning. We appreciate the feedback, which has helped us strengthen the paper.

---

### Official Review · Reviewer_MwZ1 · 2025-10-31

**Soundness:** 3
**Presentation:** 3
**Contribution:** 3
**Rating:** 6
**Confidence:** 3

**Summary:**

This paper proposes Resa, a family of reasoning models, and introduces the SAE-Tuning framework to efficiently elicit reasoning abilities in language models. SAE-Tuning first trains a sparse autoencoder (SAE) on a source model (e.g., Qwen-style Tina) using CoT-free verified question-answer data to extract latent reasoning features, then freezes the SAE and embeds it into a target model (e.g., R1-Distill) for guided supervised fine-tuning.

**Strengths:**

Experiments show the method retains ~97% performance of RL-trained models while cutting training costs by 2000x (to ~$1) and time by 450x, solving the high-cost issue of RL and CoT-based SFT while maintaining competitive performance
proving the generalizability and modularity of SAE-extracted reasoning abilities, breaking the limitation of task-specific or model-specific reasoning tuning

**Weaknesses:**

- most experiments focus on small models (1.5B/3B), leaving uncertainty about whether SAE-Tuning maintains efficiency and performance on larger models
- The method also relies on source and target models sharing the same architecture, which restricts its applicability across different model families
- the analysis of "reasoning features" remains superficial; while the paper identifies their layer-wise distribution, it lacks in-depth interpretation of what specific reasoning components these features correspond to (e.g., logical deduction vs. numerical calculation).

**Questions:**

see Weaknesses

---

> ### Author Response · Authors · 2025-11-21
>
> We thank the reviewer for the thoughtful assessment and are glad they found the method’s efficiency, competitive performance, and the demonstrated generalizability/modularity to be key strengths. We appreciate the reviewer highlighting important questions regarding scalability, architectural assumptions, and the depth of the reasoning-feature analysis. **We have updated our paper to remove confusion, improve content ordering and add new results to address these questions thoroughly (all updates in our revised paper are marked with blue text).**
>
> ### **Scalability to larger models**
>
> We agree with the importance of showing that SAE-Tuning remains effective at larger scales. To address this, we have added new experiments on larger models. In particular, we have added experiments on 8B-parameter Llama-style models (OctoThinker-8B-Base), in Table 3 of our updated draft.
>
> Concretely, SAE-Tuning improves OctoThinker-8B variants by +5 to 8.5 points on AMC23 and by up to +3.2 points on MATH500 (e.g., Hybrid: 42.60% to 45.80%). Although the strongest 8B baseline (Long variant on MATH500) leaves limited headroom, SAE-Tuning still improves AMC23 by +8.5 points for that model.
>
> We agree that exploring even larger models is an important next step. These new 8B results provide first evidence that the mechanism generalizes beyond small models while helping clarify the practical scalability of SAE-Tuning.
>
> ### **Architectural constraints**
>
> We agree with the reviewer that our current implementation requires the source and target models to share the same architecture. This arises from Stage II of SAE-Tuning, where the frozen SAE is inserted directly into the target model’s computation graph. This insertion requires matched hidden dimensions and block structure.
>
> At the same time, we aimed to validate that SAE-Tuning is not specific to a single architecture family. To this end, we applied the full pipeline end-to-end to both Qwen-style and Llama-style models (modified paper: Sections 4.1 and 4.2). Although these experiments remain within-family, they suggest that the underlying mechanism is applicable across architectures when the SAE is trained natively within each model family.
>
> We agree that true cross-architecture transfer (e.g., Qwen to Llama) is an exciting direction. As the reviewer notes, achieving this would likely require a learned mapping (e.g., a transcoder or cross-layer alignment module) that transports sparse feature spaces across different hidden dimensions.
>
> ### **Depth of "reasoning feature" analysis**
>
> We appreciate the reviewer’s request for deeper interpretability analysis. In the revised manuscript (Appendix C.2 in the modified version), we take an initial step by proposing a quantitative proxy for identifying reasoning-related SAE features—those that activate exclusively on \<think\> tokens—and by measuring their layer-wise distribution. We then show that this distribution closely tracks the performance distribution of Resa models trained with different SAE hookpoints (via a 3-component GMM analysis). This correlation provides evidence that these features capture meaningful structure related to reasoning.
>
> At the same time, we agree this analysis does not yet assign semantic interpretations to individual features (e.g., whether they correspond to algebraic manipulation, logical inference, or specific numerical subskills). Developing such fine-grained semantic characterizations remains an open challenge for SAE-based interpretability. Our methodology identifies a concrete subset of features consistently tied to reasoning performance, which we believe can serve as a targeted substrate for future interpretability work. We will clarify this distinction in the paper and highlight it as an important future research direction.
>
> We thank the reviewer again for their constructive feedback and are encouraged by their positive overall assessment.

---

> > ### Comment · Reviewer_MwZ1 · 2025-11-27
> >
> > After reviewing the other comments, I agree with Reviewer CHYk's concerns. The core theoretical claims lack sufficient evidence, which significantly weakens the paper's foundation. Furthermore, many of the statements are misleading and confusing and require substantial clarification. As a result, I am lowering my score.

---

### Author Response · Authors · 2025-12-03
**Summary of Revisions and Responses**

We thank the reviewers (MwZ1, 2vob, CHYK, CEct) for their constructive feedback, which highlighted the novelty of SAE-Tuning as a cost-effective alternative to RL for reasoning elicitation. Based on their suggestions, we have significantly revised the manuscript (changes marked in blue) to clarify our claims on efficiency, scalability, and baselines.

Below is a summary of how we addressed the key concerns, including new experimental results conducted during the discussion phase regarding Long-CoT transfer.

### **Scalability to Larger Models (Reviewers MwZ1, 2vob, CEct)**

To address concerns that our experiments were limited to small models (1.5B/3B), we’ve added latent elicitation results for 8B Llama-style models.

- *New Results (Table 3)*: SAE-Tuning improves OctoThinker-8B variants by +5 to +8.5 points on AMC23 and up to +3.2 points on MATH500, demonstrating that the mechanism generalizes to larger scales and different architectures.

### **Transferring Long-CoT to Non-Reasoning Models (Follow-up to Reviewer CEct)**

In a follow-up discussion, Reviewer CEct asked if SAE-Tuning could transfer Long-CoT capabilities from a source model (R1-Distill) to a target model that lacks inherent Long-CoT capabilities (rather than just enhancing a model that already has them).
We conducted this experiment during the rebuttal phase using Qwen2.5-Math-1.5B as the target. The results confirm successful transfer of reasoning structures:

| | AIME24 | AIME25 | AMC23 | MATH500 | GPQA | Minerva |
| :--- | :--- | :--- | :--- | :--- | :--- | :--- |
| Qwen2.5-Math-1.5B | 3.33 | 3.33 | 35 | 42.6 | 22.72 | 13.23 |
| R1-Distill-SAE-Tune-Qwen2.5-Math-1.5B | 6.67 | 3.33 | 40 | 42.4 | 30.8 | 12.5 |

This proves SAE-Tuning acts as a structural adapter, allowing us to inject reasoning patterns (Long-CoT) into models that do not natively possess them, improving GPQA performance by over 8 points.

### **Clarifying "Efficiency" and Independence from RL Teachers (Reviewers 2vob, CHYK)**

Reviewers questioned whether our "2000x efficiency" claim relied on the existence of a costly upstream RL teacher (Tina).

- *Clarification*: We clarified that SAE-Tuning does not require an RL teacher. It can function as an end-to-end replacement for the RL stage.
- *Evidence*: We highlighted Resa-STILL-v5, where we train an SAE from scratch on the base model (R1-Distill) and elicit reasoning directly into the same base model. This configuration matches the performance of the expensive RL-trained Tina-STILL (48.06% vs 48.16%) while bypassing the RL stage entirely.
- *Fair Baselines*: We clarified for Reviewer CHYK that our SFT baseline (STILL-CoT-free-SFT) used the exact same base model (R1-Distill) as our method. The baseline performance degraded (41.18% $\to$ 39.00%) due to catastrophic forgetting on CoT-free data, whereas SAE-Tuning succeeded (41.18% $\to$ 48.06%), proving the method's value.

### **Interpretability of Reasoning Features (Reviewers MwZ1, CHYK)**

We strengthened our claim regarding "reasoning features”.

- *Analysis*: We introduced a quantitative proxy for reasoning features (SAE latents that activate exclusively on <think> tokens).
- *Result*: A Gaussian Mixture Model (GMM) analysis shows that the layer-wise density of these features strongly correlates with downstream reasoning performance, providing empirical grounding for our "building blocks" hypothesis.

We believe the revised paper offers a robust, verified, and efficient methodology for reasoning elicitation. By addressing the reviewers' concerns on baselines and scalability, and by demonstrating cross-model capability transfer in our new experiments, we have strengthened the evidence that SAE-Tuning is a viable, transparent alternative to standard RL post-training.

---

### Meta-Review · Area_Chair_9BcH · 2026-01-06

**Summary:**

While reviewers acknowledged the novelty of using sparse autoencoders for reasoning transfer and the strong empirical results at small scales (Reviewers MwZ1, CHYk, and CEct), several substantive concerns informed my recommendation. Initially, several reviewers questioned the strength and evidentiary support of the paper’s core theoretical claims about "reasoning features" noting that these claims are somewhat overstated (Reviewers CHYk and 2vob). The efficiency claims were also a concern, with reviewers arguing that they were potentially misleading due to reliance on upstream RL-distilled models and unclear comparisons to standard distillation baselines (Reviewers 2vob, CHYk). While the authors conducted additional experiments and aimed to provide some clarifications during the rebuttal period, I feel these are substantial updates to the original paper and necessitate another round of reviewing with an updated/revised work for fair evaluation. Additionally, in my opinion, as the authors also agree, concerns about scalability are somewhat unresolved as well, including the practicality of large SAEs for larger models and limited evidence beyond math-centric tasks and relatively small model sizes, even after adding 8B param model experiments (Reviewers MwZ1, 2vob, and CEct). Although the rebuttal addressed a number of clarity and experimental concerns, the remaining issues around theoretical grounding, efficiency, and overall framing ultimately hinder acceptance.

**Reviewer Concerns:**

The rebuttal strengthened the paper as the authors provided clarifications/experiments to alleviate concerns raised by the reviewers. In my opinion, issues around clarity of baselines and experimental fairness were mostly resolved, particularly the confusion about the STILL-CoT-free-SFT baseline and base model performance (Reviewers CHYk, 2vob). Concerns about scalability beyond very small models were somewhat partially addressed through added experiments on 8B Llama-style models (Reviewers MwZ1 and CEct).

However, several significant concerns still remain outstanding. I agree with reviewers CHYk and MwZ1 about whether the core theoretical premise is sufficiently justified, as the new analyses remain largely correlational. The authors agree that they overstated their original claims and decided to appropriately soften them based on feedback. These as well as the efficiency claims, while somewhat clarified, still require an updation and substantial revisions to the original paper and its framing. Finally, scalability concerns persist regarding the size and practicality of SAEs for much larger models and more general task domains, which are acknowledged but not convincingly resolved in my view (in the current work at least).

**Reviewer Scores:**

- Reviewer MwZ1: I believe they would have likely reduced their score as indicated in their response to authors (owing to theoretical grounding issues and need for substantial paper revision).
- Reviewer 2vob: I believe they would have maintained their score owing to concerns around the efficiency claim and paper revision/framing.
- Reviewer CHYk: I believe they would have likely reduced their score due to outstanding concerns re: theoretical claims, among other issues.
- Reviewer CEct: The reviewer would have maintained their positive score or increased it as they stated to authors.

Overall the paper has merit, and outstanding concerns can be improved upon in a revision/resubmission.

---

### Decision · Program_Chairs · 2026-01-26

Reject